# Ion occupancy of the selectivity filter controls opening of a cytoplasmic gate in the K2P channel TALK-2

Lea C. Neelsen[1,7], Elena B. Riel [1,2,7], Susanne Rinné [3,7], Freya-Rebecca Schmid[2], Björn C. Jürs [1,4], Mauricio Bedoya [5,6], Jan P. Langer[1], Bisher Eymsh[1], Aytug K. Kiper [3], Sönke Cordeiro[1], Niels Decher [3,8] ✉, Thomas Baukrowitz [1,8] ✉ & Marcus Schewe [1,8] ✉

Two-pore domain K+ (K2P) channel activity was previously thought to be controlled primarily via a selectivity filter (SF) gate. However, recent crystal structures of TASK-1 and TASK-2 revealed a lower gate at the cytoplasmic pore entrance. Here, we report functional evidence of such a lower gate in the K2P channel K2P17.1 (TALK-2, TASK-4). We identified compounds (drugs and lipids) and mutations that opened the lower gate allowing the fast modification of pore cysteine residues. Surprisingly, stimuli that directly target the SF gate (i.e., pHe., Rb+ permeation, membrane depolarization) also opened the cytoplasmic gate. Reciprocally, opening of the lower gate reduced the electric work to open the SF via voltage driven ion binding. Therefore, it appears that the SF is so rigidly locked into the TALK-2 protein structure that changes in ion occupancy can pry open a distant lower gate and, vice versa, opening of the lower gate concurrently promote SF gate opening. This concept might extent to other K+ channels that contain two gates (e.g., voltage-gated K+ channels) for which such a positive gate coupling has been suggested, but so far not directly demonstrated.

TWIK-related alkaline-pH-activated potassium (TALK-2, K2P17.1, *KCNK17*) channels are members of the two-pore domain K+ (K2P) channel family. They were also referred to as TWIK-related acid-sensitive potassium (TASK-4) channels when first identified in 2001[1]. TALK-2 channels are expressed in various human cell types and organs (i.e., pancreas, aorta, brain, liver, placenta, and heart) of the human body[1–3]. Despite their widespread distribution the functional role of these channels in biological processes has not been established yet, especially in contrast to other well-investigated acid-sensitive K2P

channels such as TASK-1[4–9]. However, its malfunction, up- or down-regulation or genetic variants of TALK-2 K2P channels have been associated with a number of cardiovascular diseases such as cardiac conduction disorders[10], ischemic stroke, and cerebral hemorrhage[11–14] as well as arrhythmias including atrial fibrillation, idiopathic ventricular fibrillation and long QT syndrome[10,15,16]. Furthermore, TALK-2 channels are highly and specifically expressed in the human pancreas and are considered as a risk factor for the pathogenesis of type 2 diabetes[17,18]. TALK-2 channels are activated by alkaline extracellular pH

[1]Institute of Physiology, Christian-Albrechts University of Kiel, Kiel, Germany. [2]Department of Anesthesiology, Weill Cornell Medical College, New York, NY, USA. [3]Institute of Physiology and Pathophysiology, Philipps-University of Marburg, Marburg, Germany. [4]MSH Medical School Hamburg, University of Applied Sciences and Medical University, Hamburg, Germany. [5]Centro de Investigación de Estudios Avanzados del Maule (CIEAM), Vicerrectoría de Investigación y Postgrado, Universidad Católica del Maule, Talca, Chile. [6]Laboratorio de Bioinformática y Química Computacional (LBQC), Departamento de Medicina Traslacional, Facultad de Medicina, Universidad Católica del Maule, Talca, Chile. [7]These authors contributed equally: Lea C. Neelsen, Elena B. Riel, Susanne Rinné. [8]These authors jointly supervised this work: Niels Decher, Thomas Baukrowitz, Marcus Schewe. ✉e-mail: decher@staff.uni-marburg.de; t.baukrowitz@physiologie.uni-kiel.de; m.schewe@physiologie.uni-kiel.de

(pH$_e$ > 7.4) that is thought to occur by deprotonation of an extracellular lysine causing the opening of the SF gate[1,2,18,19]. Furthermore, TALK-2 channel currents are enhanced by the production of nitric oxide radicals and reactive oxygen species[18]. Like most K$_{2P}$ channels, TALK-2 channels are sensitive to changes in membrane voltage and permeating ion species[20]. Recently, polyanionic lipids of the fatty acid metabolism (e.g. oleoyl-CoA) have been identified as natural TALK-2 channel ligands, increasing channel activity by more than 100-fold[21]. How exactly these stimuli regulate the opening and closing of TALK-2 K$_{2P}$ channels is, with the exception of voltage and pH$_e$ acting at the SF gate, so far unclear. TALK-2 channels are functional dimers and exhibit – as all other 14 K$_{2P}$ channel family members - a characteristic topology of four transmembrane domains (TM1 to TM4) and two pore-forming domains (P1 and P2) within each channel subunit. The two P1 and P2 domains form the pseudo-tetrameric selectivity filter (SF) of the channel upon dimerization[22–24]. For the last two decades, K$_{2P}$ channels were thought to be gated at the SF and, thus, the various physicochemical stimuli acting on different regions of the channel (in particular on the cytoplasmic C-terminus) finally converge on the primary filter gate[20,25–29]. Surprisingly, the recently resolved structures of TASK-1 and TASK-2, identified additional inner/cytoplasmic gates (further referred to as lower gates)[30,31]. In TASK-1 this gate is formed by the crossing (therefore termed X-gate) of the late TM4 domains[31], however, an activation mechanism for this gate is currently unknown. Based on the two cryo-EM structures of TASK-2 generated at pH 8.5 (open channel) and pH 6.5 (closed channel) the lower gate of TASK-2 is mainly formed by the interaction of two corresponding TM4 residues (K245, N243), hypothesized to function as a molecular barrier in the process of pH gating[30]. In this study, we employ cysteine modification, alanine scanning mutagenesis, homology modeling and various pore blockers to identify and characterize an additional lower gate in TALK-2 channels. These approaches provide information on the status of the SF gate and lower gate, respectively, and reveal that the two gates are strongly positively-coupled. We show that the ion occupancy of the SF controls opening of the lower gate and estimate the electrical work required to open the lower gate. Our results establish a strong positive coupling of the two gates that can be envisioned as a concerted structural change involving both gates. Further, we demonstrate that the lower gate produces a state-dependent pharmacology that is unique in K$_{2P}$ channels.

## Results

### Probing for a cytoplasmic constriction in TALK-2 channels with cysteine modification

The basal activity of TALK-2 channels in excised patches is low, but pharmacological compounds (e.g. 2-APB)[32] or polyanionic lipids (e.g., oleoyl-CoA)[21] can induce large TALK-2 channel currents (Fig. 1a, f). However, the binding sites of these compounds and the mechanisms through which they open the ion permeation pathway are currently unknown. To investigate the latter, we utilized a cysteine modification assay. Control experiments ensured that WT TALK-2 currents were insensitive to the application of the sulfhydryl reactive compound (2-(Trimethylammonium)ethyl) MethaneThioSulfonate (MTS-ET) regardless whether applied on the low-activity basal state (Supplementary Fig. 1a, Supplementary Table 1) or the high activity state induced with e.g., 2-APB or oleoyl-CoA (Supplementary Fig. 1b, d, f). To test for a cytoplasmic constriction in TALK-2, we introduced a cysteine at amino acid position 145 (L145C) in TM2, that corresponds to a residue in TREK-1 (G186C) located in the pore cavity below the SF and previously shown to result in a permeation block upon cysteine modification in TREK-1[29]. TALK-2 L145C mutant channels showed low basal activity, and both 2-APB and oleoyl-CoA produced robust activation very similar to the WT (Fig. 1g and Supplementary Figs. 1c, e, g and 5a). Application of MTS-ET on the low activity basal state had no effect on the channel activity (Fig. 1d left panel). In marked contrast,

application of MTS-ET on L145C TALK-2 currents activated by 2-APB or oleoyl-CoA resulted in complete and irreversible current inhibition, indicating the chemical modification of L145C (Fig. 1d middle and right panel, Supplementary Table 1). These findings indicate that access of MTS-ET to L145C is blocked in the low-activity state of the channel but possible upon activation. To gain a better structural understanding, we generated TALK-2 homology models based on the TASK-1 and TASK-2 structures that both show a lower permeation constriction (see "Methods" section)[30,31]. In these models, the side chain of L145 points into the permeation pathway at a position between the SF and the lower gates (Fig. 1b, c and Supplementary Fig. 2a). Furthermore, we used these TALK-2 models to pick a residue for cysteine substitution (Q266) pointing into the cytosol directly below the lower constrictions. Application of MTS-ET to Q266C TALK-2 mutant channels caused a mono-exponential and irreversible current activation (Fig. 1e left panel). Importantly, the observed modification occurred at a similar rate in both the low- and high-activity states suggesting similar access to the cysteine under both conditions (Fig. 1e and Supplementary Table 2). The fact that MTS-ET modification activated Q266C TALK-2 mutant channels might indicate that the modification at this position (i.e., close to the putative lower gate) destabilizes the closed gate and thus results in channel activation. Our findings suggest the existence of a lower constriction blocking MTS-ET access at the level of the X-gate in TASK-1 that is opened by 2-APB or oleoyl-CoA in TALK-2 channels. Accordingly, we observed that current activation and the rate of L145C modification concurrently increased with the 2-APB concentration levelling off at a concentration of 2.0 mM 2-APB that caused maximal current activation (Fig. 1f–h and Supplementary Table 1).

### The lower constriction functions as a permeation gate

Our data strongly suggest a gated lower pore constriction site. However, whether this constriction is an actual permeation gate is still disputable, as 2-APB and oleoyl-CoA could have also opened the SF gate to cause activation. Furthermore, a constriction that blocks MTS-ET access may not necessarily block ion permeation. Our attempts to use silver ions (Ag$^+$, which is much smaller than MTS-ET and similar in size to K$^+$) to probe for a permeation gate were not conclusive as Ag$^+$ also inhibited WT TALK-2 channels. Therefore, we addressed this issue with a different approach by taking advantage of the fact that Rb$^+$ has a strong activating effect on the TALK-2 SF[20] and to do so, Rb$^+$ needs to pass the lower constriction. And indeed, a stepwise increase in the 2-APB concentration resulted in a stepwise increase in Rb$^+$ activation as if 2-APB activation had removed a constriction that prevented Rb$^+$ to access the SF (Fig. 2a, c). Accordingly, the same effect on Rb$^+$ activation was also observed when TALK-2 was activated by oleoyl-CoA (Supplementary Fig. 3a). As a control, we performed the same experiment with TREK-1 channels, since these K$_{2P}$ channels lack a lower gate but also undergo activation upon 2-APB application (Fig. 1g). In TREK-1, Rb$^+$ activation was the strongest in the absence of 2-APB activation, indicating that Rb$^+$ ions had free access to the SF in the low activity state of the channel (Fig. 2b, c). Moreover, 2-APB activation progressively covered up Rb$^+$ activation consistent with the concept that TREK-1 channels lack a lower gate and both activating stimuli (2-APB and Rb$^+$) converge onto the SF gate (Fig. 2b). These results suggest that the lower constriction in TALK-2 channels functions as a permeation gate that must be open in addition to the SF gate to allow ion conduction (Fig. 2d).

### Characterization of the lower gate by scanning mutagenesis and single-channel recordings

We performed systematic alanine scanning mutagenesis in the TM regions that contain the lower gates in TASK-1 and TASK-2 (i.e., TM2 and TM4) to identify mutations that would affect the stability of the lower gate in TALK-2 (Fig. 3a, b and Supplementary Fig. 4a). This

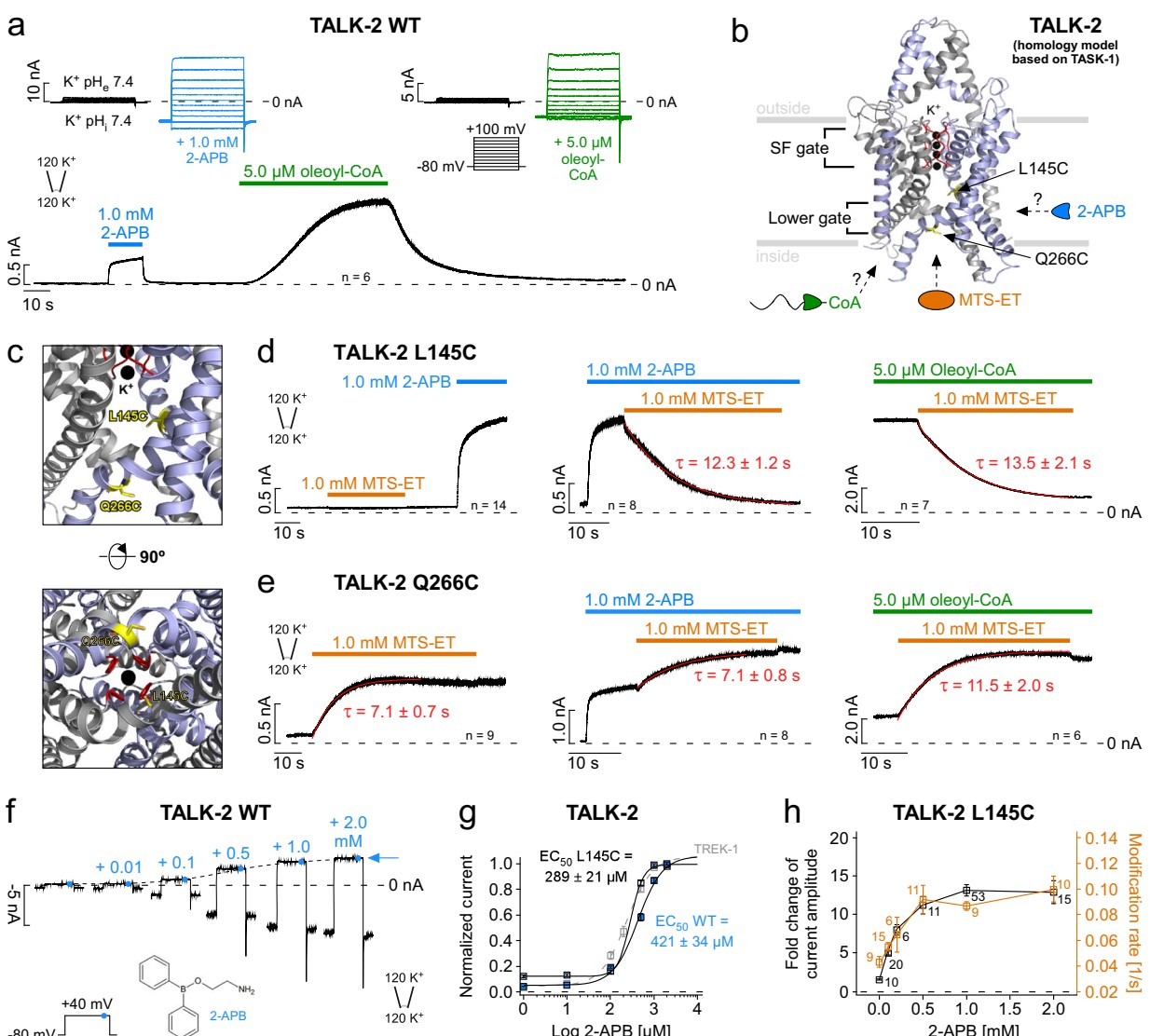

**Fig. 1 | State-dependent modification of inner pore cysteine residues in TALK-2 $K_{2P}$ channels. a** Representative current trace measured at +40 mV from an inside-out patch expressing WT TALK-2 channels with symmetrical $K^+$ concentrations at pH 7.4. Channel currents were activated with the indicated compounds (1.0 mM 2-APB and 5.0 μM oleoyl-CoA) applied to the intracellular membrane side. Inlays show current-voltage responses of 2-APB- (blue) and oleoyl-CoA-activated (green) channels compared to basal state (black) using the indicated voltage step protocol. **b** Pore homology model of TALK-2 based on the crystal structure of TASK-1 (PDB ID: 6RV3, chains A, B) with the SF highlighted red, $K^+$ ions black, and introduced cysteine residues (L145C and Q266C) for MTS-ET modification yellow. **c** Pore cavity zoom-in displaying the localization of L145C in the inner cavity and Q266C at the intracellular end of the pore. **d** Representative measurement of TALK-2 L145C channels showing state-dependent MTS-ET modification with no effect under unstimulated (basal) conditions or inhibition upon application of 1.0 mM MTS-ET in pre-activated states with 1.0 mM 2-APB (blue) or 5.0 μM oleoyl-CoA (green) with the indicated time constants (τ), respectively. **e** Measurement as in (d) with TALK-2 Q266C channels showing state-independent modification with activation upon application of 1.0 mM MTS-ET. **f** Current responses recorded using the indicated voltage step protocol in symmetrical $K^+$ showing activation of WT TALK-2 with increasing 2-APB concentrations. The dotted line shows the increase and saturation of current amplitudes with 2-APB at + 40 mV. **g** 2-APB dose-response curves analyzed from measurements as in **f** for WT TALK-2 (blue, $n = 26$), TALK-2 L145C (black, $n = 7$), and WT TREK-1 (gray, $n = 28$) channels. **h** Correlation between the fold change in current amplitudes of TALK-2 L145C channels at +40 mV (black squares) and the rate of MTS-ET modification (1/τ) at +40 mV (orange squares) with different 2-APB concentrations. Data shown are the mean ± s.e.m and the number (n) of independent experiments is indicated in the figure and supplementary tables 1 and 2. The representative experiments were repeated with the similar results as indicated in the figure.

approach identified four gain of function (g-o-f) mutations (V146A in TM2, W255A, L262A and L264A in TM4) that markedly increased TALK-2 basal currents 31.9 ± 3.4-fold (V146A), 4,5 ± 0.2-fold (W255A), 25.1 ± 2.1-fold (L262A), and 93.3 ± 9.8-fold (L264A) as assessed in two-electrode voltage-clamp (TEVC) experiments with oocytes (Fig. 3a). Mapping of the g-o-f residues on our TALK-2 homology models showed a clustering of the g-o-f residues in the region of the lower pore constrictions identified in TASK-1 and TASK-2 at the cytosolic pore entrance of the channels (Fig. 3c, Supplementary Fig. 4b). In the TASK-1-based TALK-2 homology model the two residues showing the strongest g-o-f phenotype (i.e., V146A and L264A) are in direct proximity and could form a permeation constriction (Fig. 3c). Further, the g-o-f mutant channel with the strongest effect (i.e., L264A) and WT TALK-2 channels were further analyzed with inside-out single channel recordings (Fig. 3d–h). As expected, the basal single channel activity of WT TALK-2 was very low with a relative channel-open probability ($NP_O$) of 3.9 ± 0.2 % and the L264A mutation resulted in a large increase to 46.7 ± 12.1 %, while the single-channel amplitude was not affected

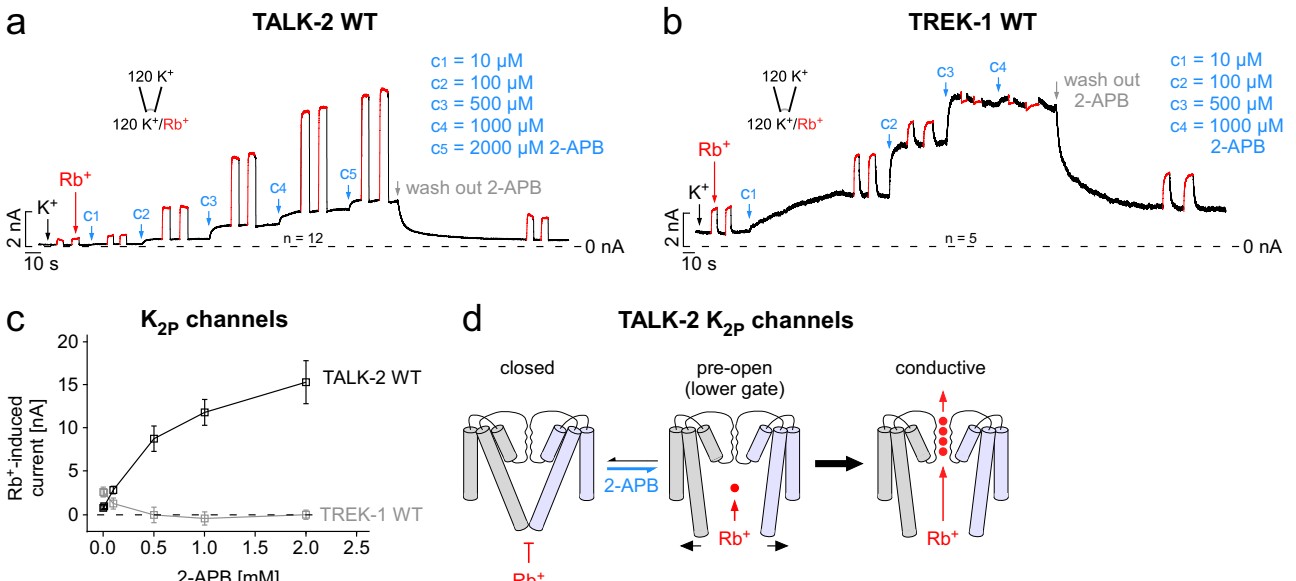

**Fig. 2 | The lower constriction functions as a permeation gate. a** Representative measurement of WT TALK-2 channels from an inside-out patch in symmetrical $K^+$ at +40 mV with increasing 2-APB concentrations ($c_1$–$c_5$) applied from the intracellular side at indicated time points (blue arrows). At steady-state, current levels with 2-APB intracellular $K^+$ was exchanged by $Rb^+$ showing an enhanced activatory $Rb^+$ ion effect on the SF in 2-APB pre-activated channels. **b** Recording as in **a** for WT TREK-1 $K_{2P}$ channels showing the stepwise loss of $Rb^+$ activation in the presence of increasing 2-APB concentrations. **c** Correlation of $Rb^+$-induced currents from measurements as in **a**, **b** in the presence of 0.01, 0.1, 0.5, 1.0, or 2.0 mM 2-APB for either WT TALK-2 (black, $n = 12$) or WT TREK-1 (gray, $n = 5$) channels. **d** Gating scheme highlighting the effect of 2-APB and $Rb^+$ on the lower and selectivity filter gate in TALK-2 channels. Data shown are the mean ± s.e.m and the number ($n$) of repeats of the representative measurements with similar results is indicated in the figure.

(Fig. 3d, e). The increase in $NP_O$ appeared to primarily result from a destabilization of the closed state, as we observed a strong shortening of the short (-13-fold) and long closed times (-16-fold) (Fig. 3f), whereas the open times were comparatively little (-2-fold) affected (Fig. 3g). Overall, these changes result in a 29-fold increase of channel activity spent in a burst type mode with $767 \pm 123$ bursts/min for L264A channels compared to $27 \pm 17$ burst/min for WT channels (Fig. 3h). The observed shortening in closed times is consistent with the concept that the g-o-f mutations destabilize a permeation gate that now opens much more frequently. Further, if this destabilized permeation gate corresponds to the lower gate, we expect fast modification of the L145C mutation as indeed observed in L264A/L145C double mutant channels (Fig. 3i, j). Moreover, all four g-o-f mutants resulted in fast L145C modification (in the absence of ligand activation) (Fig. 3i–k) with rates that roughly correlated to the g-o-f effect (Fig. 3a) with L264A showing the fastest modification (Fig. 3i, k and Supplementary Table 2).

**Gate coupling in TALK-2 channels**

The existence of two activation gates in TALK-2 raises the question if they are coupled. To address this question, we tested a stimulus that directly affects the SF gate. Extracellular alkalinization is thought to open the SF gate in TALK-2 by deprotonation of a lysine residue (K242) located at the outer pore helix connected to the SF[19] (Fig. 4b). Accordingly, raising the $pH_e$ from 7.4 to 9.5 resulted in large TALK-2 currents (Fig. 4a and Supplementary Fig. 5a, b). Under this condition, we determined the MTS-ET accessibility of L145C and, surprisingly, observed a similar fast modification rate as seen with maximal (2.0 mM) 2-APB activation at $pH_e$ 7.4 (Fig. 4c and Supplementary Table 1). However, raising the $pH_e$ to 9.5 had little effect on the modification of Q266C consistent with the localization of this position at the cytoplasm-facing side of lower gate (Supplementary Fig. 5c and Supplementary Table 2). These results imply that opening the SF gate by high $pH_e$ had also opened the lower gate in TALK-2.

**Ion occupancy of the SF controls the opening of the lower gate**

We have previously shown that the SF acts as an ion-flux voltage gate in many $K_{2P}$ channels including TALK-2[20]. According to this concept, the low basal activity of exclusively SF-gated $K_{2P}$ channels – such as TREK-1 – results from an inactivated and ion-depleted filter. Upon depolarization, ions are forced into the SF by the transmembrane electric field to induce voltage-dependent channel activation. The electrical work necessary to open the SF is reflected in the conductance-voltage ($G$–$V$) curves obtained by plotting the relative open probability (tail current amplitudes) against the membrane pre-pulse voltage (Fig. 4d, e). Particularly strong ion activation is seen with intracellular $Rb^+$ as this ion appears to stabilize the conductive state of the filter more efficiently than $K^+$ (Fig. 4g, h and Supplementary Fig. 5d, e). By monitoring the modification of L145C mutant channels we tested whether opening of the SF with voltage and $Rb^+$ would also affect the status of the lower gate (Fig. 4f, h, i). Indeed, activation of TALK-2 channels upon $K^+$ by $Rb^+$ replacement at a membrane potential of +40 mV allowed strong L145C MTS-ET modification (Fig. 4h and Supplementary Table 1). Thus, the mere change in SF ion occupancy upon exchanging $K^+$ by $Rb^+$ in the SF is sufficient to markedly increase the $P_O$ of the lower gate allowing L145C modification (Fig. 4f). As expected, the L145C modification rate was strongly voltage-dependent and the increase in channel $P_O$ reflected in the $G$–$V$ curve mirrored the increase in modification rate and, thus, voltage-dependent opening of the SF gate concurrently opened also the lower gate (Fig. 4e, i).

**The properties of the lower gate resulted in a state-dependent TALK-2 pharmacology**

In voltage-gated $K^+$ ($K_v$) channels, the closing of the lower gate can be prevented by the binding of blockers such as tetra-pentyl-ammonium (TPenA) in the pore cavity and, thereby, causing a slowing of the deactivation kinetics which is known as the 'foot in the door' effect[33,34]. To test for such a mechanism in TALK-2, we activated the channels with a voltage step to +100 mV in the presence of intracellular $Rb^+$ followed

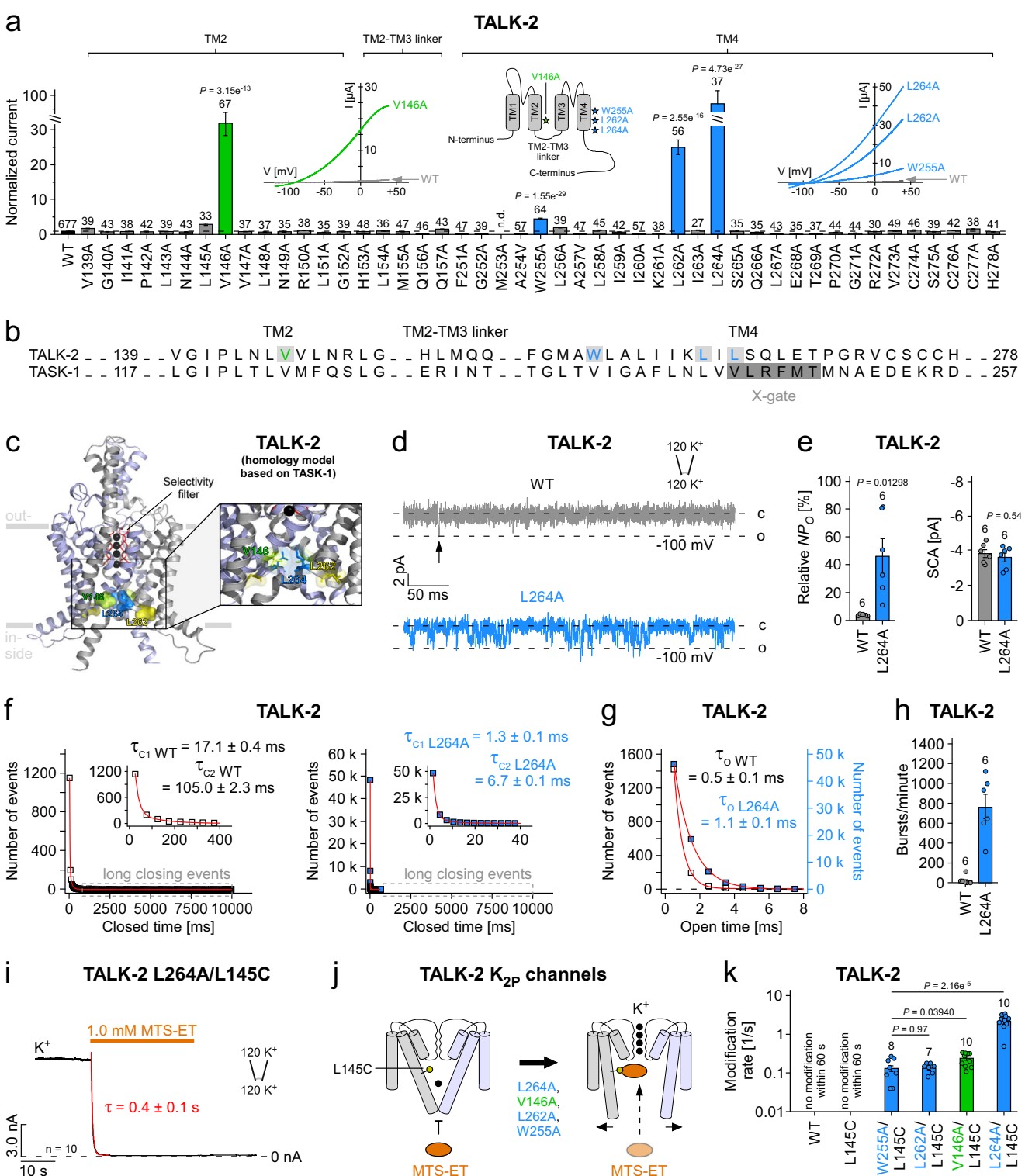

by a repolarization step to -80 mV (to induce tail currents) with and without the K$_{2P}$ channel pore blocker TPenA. Remarkably, we observed a tail current cross-over indicating that the deactivation time course is slowed by TPenA inhibition; i.e., with ~50 % TPenA block of the tail current amplitudes we observed a ~2.5-fold slowing of the deactivation kinetics (Fig. 5a). We hypothesize, that the two gates are strongly coupled and, therefore, opening the SF gate by depolarization should also open the lower gate, which then consequently would allow blocker (like TPenA) binding within the pore cavity (Fig. 5a cartoon). Upon repolarization, however, the bound TPenA blocker obstructs closure of the

lower gate and thereby, hinders the SF from closure/inactivation (gate coupling). In accordance, when we tested this protocol on TREK-2 K$_{2P}$ channels, known to lack a lower gate, TPenA inhibition had no effect on the tail current kinetics indicating that here the SF gate is able to close unhindered with the pore blocker bound (Fig. 5b). Further, TPenA inhibition had likewise little effect on tail current kinetics of 2-APB-activated TALK-2 channels indicating that 2-APB preferentially opens the lower gate in TALK-2 (Supplementary Fig. 6a).

These findings suggest that TPenA is a state-dependent blocker in TALK-2 channels and, thereby, the apparent affinity for TPenA should

**Fig. 3 | Functional characterization of the lower gate in TALK-2 K₂ₚ channels.**
**a** Relative current amplitudes from TEVC measurements at pH 8.5 of WT and mutant TALK-2 channels. Currents were elucidated with a voltage protocol ramped from -120 mV to +45 mV within 3.5 s, analyzed at +40 mV and normalized to WT. Inlays showing representative WT TALK-2 (gray traces), TALK-2 L264A, L262A, W255A and V146A mutant channel currents (blue and green traces), respectively and a topology model of a channel protomer highlighting the localization of the g-o-f mutations. **b** Sequence alignment of the TM2, TM2-TM3 linker, and TM4 regions of the human K₂ₚ channels TALK-2 and TASK-1. **c** Pore homology model of TALK-2 based on the crystal structure of TASK-1 (PDB ID: 6RV3, chains A, B) highlighting the cluster of g-o-f mutations (V146A, L262A and L264A) at the cytosolic pore entrance. **d** Representative inside-out single channel measurements of WT TALK-2 (gray trace) and TALK-2 L264A mutant channels (blue trace) at -100 mV. **e** Relative open probability ($NP_O$) and single channel amplitudes (SCA) analyzed from recordings as

in **d** for WT and L264A TALK-2 channels ($n = 6$). **f**–**h** Analysis of the mean channel-open times (**g**), closed time events (**f**), and burst behavior (**h**; see "Methods" section) for WT and TALK-2 L264A mutant channels. **i** Representative measurement of TALK-2 L264A mutant channels additionally carrying the inner pore mutation L145C (TALK-2 L264A/L145C) at +40 mV showing a fast and irreversible modification and subsequent block upon application of 1.0 mM MTS-ET. The experiment was repeated with similar results ($n = 10$). **j** Cartoon illustrating the pore accessibility of MTS-ET in L145C mutant TALK-2 channels with or without carrying an additional g-o-f mutation. **k** Modification rates of WT, L145C and double mutant TALK-2 channels at +40 mV as indicated. Data shown are the mean ± s.e.m and the number ($n$) of independent experiments is indicated in the figure and supplementary table 2. Statistical relevance has been evaluated using unpaired, two-sided $t$-test and exact $P$ values are indicated in the figure. n.d. not determinable.

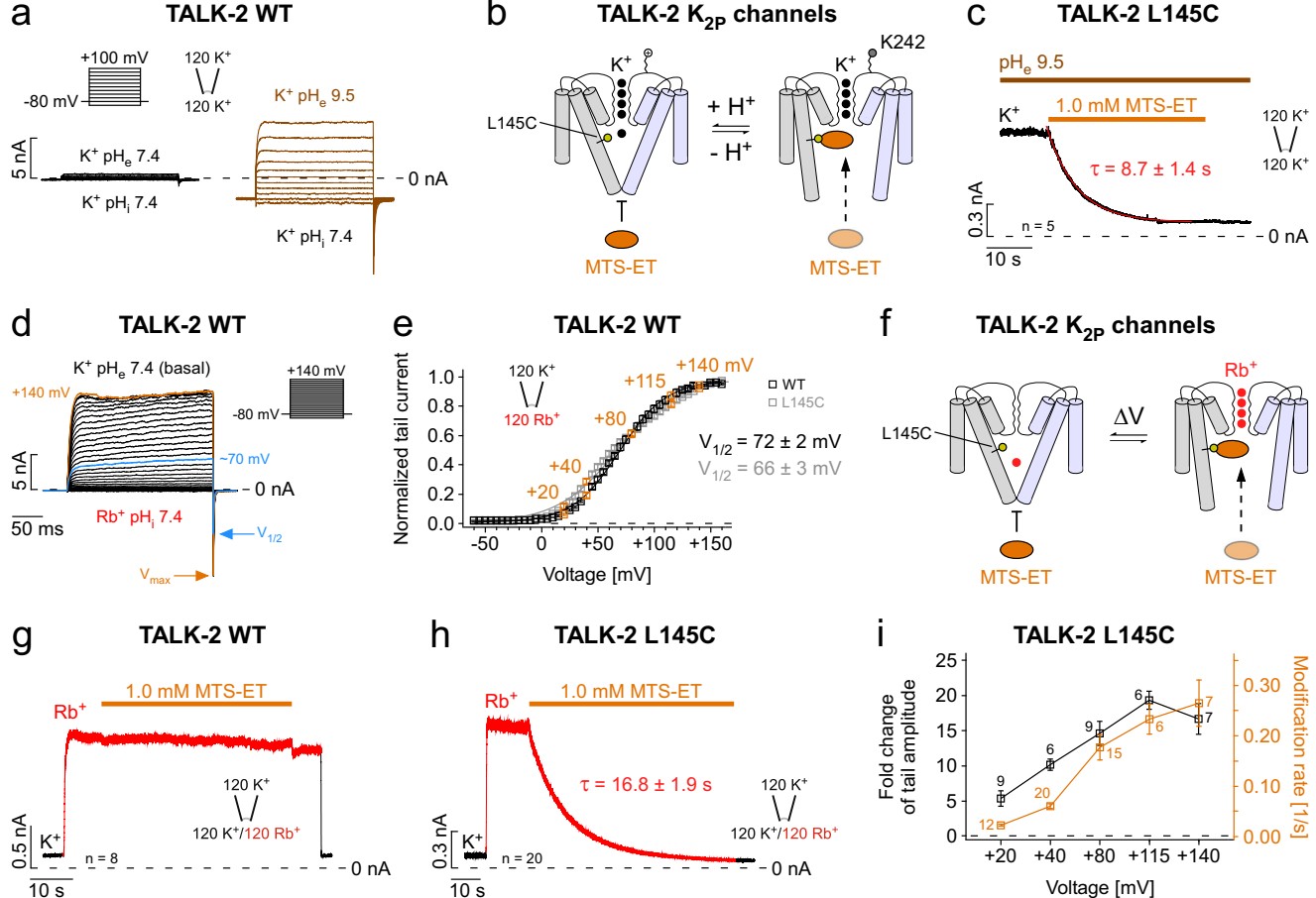

**Fig. 4 | Direct stimulation of the SF produces a state-dependent cysteine modification in the pore of TALK-2. a** TALK-2 current responses to voltage step families as indicated using symmetrical K⁺ concentrations (120 mM [K⁺]ₑₓ/120 mM [K⁺]ᵢₙₜ) at pH 7.4 on both sides (black traces), or at extracellular pH 9.5 (brown traces). **b** Cartoon illustrating a simplified TALK-2 channel gating model and pore accessibility to MTS-ET by alterations of the pHₑ that directly affects the SF. **c** Representative modification and subsequent irreversible inhibition with 1.0 mM MTS-ET of TALK-2 L145C channels pre-activated by extracellular alkalinization (pHₑ 9.5). **d** TALK-2 channel currents with intracellular Rb⁺ (120 mM [K⁺]ₑₓ/120 mM [Rb⁺]ᵢₙₜ) at pH 7.4 for different potentials as indicated showing a maximum $P_O$ reached for potentials positive to ~+135 mV (Vₘₐₓ), as further depolarizations do not increase the tail current amplitudes. **e** Voltage activation (conductance-voltage (G–V) curves) with V₁/₂ values of 72 ± 2 mV and 66 ± 3 mV of WT TALK-2 ($n = 15$) and

L145C mutant channels ($n = 10$), respectively. The highlighted voltages (orange) represent the voltage activation levels for MTS-ET modification experiments shown in **i**. **f** Cartoon of a simplified gating model with Rb⁺ as an amplifier for voltage activation targeting the SF and subsequently the lower gate in TALK-2 channels. **g**, **h** Representative measurements at +40 mV of WT (**g**) and L145C mutant TALK-2 channels (**h**) showing a non-modifiable state or an almost complete modification/ inhibition with 1.0 mM MTS-ET within 60 s in intracellular Rb⁺, respectively. **i** Correlation between the fold change of tail current amplitudes (black squares) of TALK-2 L145C channels and the incidental rate of MTS-ET modification (1/τ) (orange squares) with intracellular Rb⁺ at different potentials as indicated. Data shown are the mean ± s.e.m and the number ($n$) of independent experiments and repeats of representative measurements with similar results is indicated in the figure and supplementary tables 1–3.

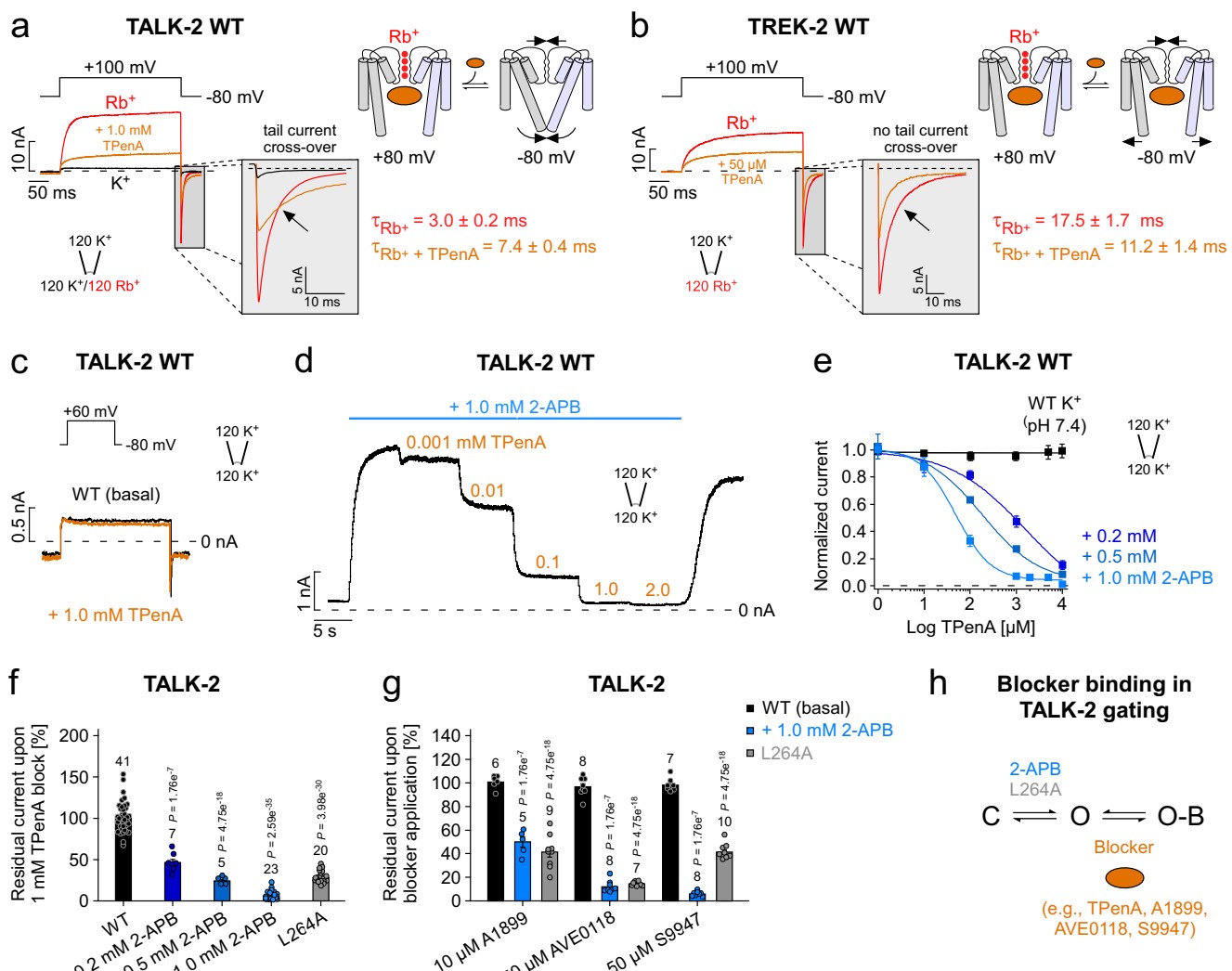

**Fig. 5 | Open channel blocker show state-dependent pore accessibility and slowing of deactivation kinetics in TALK-2. a** Current responses of WT TALK-2 channels activated with indicated voltage steps under symmetrical ion conditions with either intracellular $K^+$ (black trace, basal state) or $Rb^+$ (red trace, activated state) and with 1.0 mM TPenA in $Rb^+$ (orange trace). Note, the presence of TPenA shows slowing of deactivation resulting in a tail current cross-over. Cartoon depicting a simple model for TALK-2 channel gating, whereby $Rb^+$ activation of the SF enables blocker (e.g., TPenA) binding in the pore and unbinding facilitates lower and SF gate closure at −80 mV. **b** Same recording as in (a) with TREK-2 channels showing inhibition with 50 μM TPenA without tail current cross-over. **c** Representative current responses of WT TALK-2 channels to voltage steps as indicated in the absence (black) and presence of 1.0 mM TPenA (orang) applied to the intracellular membrane side. **d** Representative measurement of TALK-2 channel currents at +40 mV showing dose-dependent TPenA inhibition in the pre-activated

state with 1.0 mM 2-APB. **e** Dose-response curves of TPenA inhibition from measurements as in **d** for TALK-2 in unstimulated conditions (black, $n = 16$) and pre-activated states with 2-APB (blue) with altering apparent affinities for TPenA ($IC_{50}$ (0.2 mM 2-APB, $n = 7$) = 778 ± 116, $IC_{50}$ (0.5 mM 2-APB, $n = 5$) = 215 ± 28, $IC_{50}$ (1.0 mM 2-APB, $n = 16$) = 54 ± 10). **f** Residual currents of WT and L264A mutant TALK-2 channels at +40 mV upon 1.0 mM TPenA block at indicated conditions. **g** Residual currents of unstimulated (black), 2-APB pre-activated WT (blue) and L264A mutant (gray) TALK-2 channels after inhibition with the indicated blocker. **h** Simplified gating scheme indicating that blocker interact with the open state of TALK-2 to produce inhibition. Data shown are the mean ± s.e.m and the number (n) of independent experiments and repeats of representative measurements with similar results is indicated in the figure. Statistical relevance has been evaluated using unpaired, two-sided $t$-test and exact $P$ values are indicated in the figure.

strongly depend on the fraction of channels being open (Fig. 5h). Indeed, in the basal state (low $P_O$) hardly any TPenA block was observed (Fig. 5c), whereas 2-APB activation induced dose-dependent TPenA inhibition (Fig. 5d–f). Accordingly, the apparent affinity for TPenA inhibition increased dramatically with the degree of 2-APB activation ($IC_{50}$ 0.2 mM 2-APB = 1108 ± 411 μM, $IC_{50}$ 0.5 mM 2-APB = 225 ± 25 μM, $IC_{50}$ 1.0 mM 2-APB = 54 ± 10 μM) (Fig. 5e). We further explored this observation by testing several other compounds known to block $K_{2P}$ channels (e.g., TASK-1) such as A1899, AVE0118 and S9947[35]. Remarkably, for each blocker tested little inhibition was seen for unstimulated TALK-2 channels while strong inhibition was observed upon activation by 2-APB (Fig. 5g). Furthermore, the g-o-f

mutation L264A resulted in TALK-2 channels with high sensitivity to inhibition for all tested blockers without 2-APB activation further indicating that the mutation promoted opening of the lower gate (Fig. 5f, g and Supplementary Fig. 6b, c). In conclusion, the existence of the lower gate in TALK-2 transformed the state-independent pore block as seen in TREK-2 into a state-dependent (foot in the door-like) block as typical for $K_v$ channels.

**Opening of the lower gate reduces the mechanical load coupled to the voltage-powered SF gate**

The voltage-dependent activation mechanism of the SF gate in $K_{2P}$ channels allows to estimate the electrical work ($\Delta G = zF\Delta V_{1/2}$) required

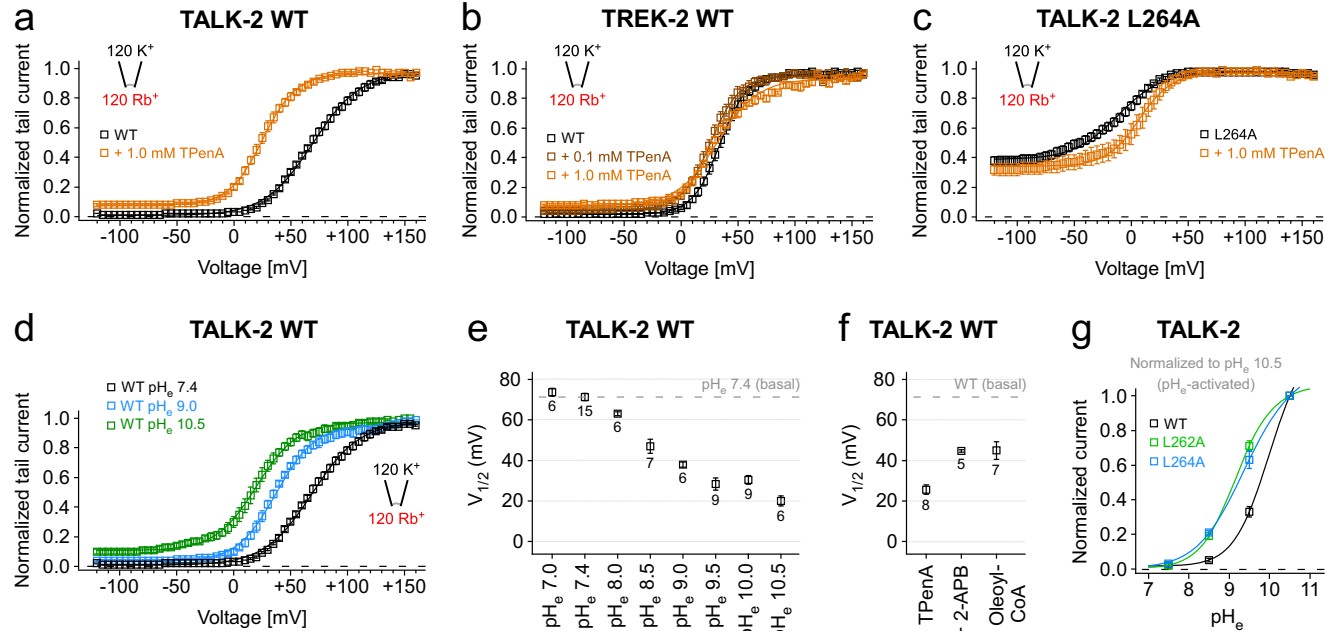

**Fig. 6 | Impact of ligand modulation on SF energetics in TALK-2 $K_{2P}$ channels.**
**a**–**c** G–V curves analyzed from current-voltage families (−120 mV to +160 mV with 5 mV increments) measured under symmetrical ion conditions with intracellular Rb⁺ of WT TALK-2 (**a**), WT TREK-2 (**b**) and L264A mutant TALK-2 channels (**c**) in the absence (black traces, $n = 15$ TALK-2, $n = 6$ TREK-2, $n = 15$ TALK-2 L264A) and presence of 0.1 mM (brown trace, $n = 7$ TREK-2) or 1.0 mM TPenA (orange traces, $n = 8$ TALK-2, $n = 7$ TREK-2, $n = 8$ TALK-2 L264A), respectively. **d** G–V curves analyzed from WT TALK-2 tail currents in the presence of $pH_e$ 7.4 (black trace, $n = 15$), pH 9.0 (blue trace, $n = 6$), and pH 10.5 (green trace, $n = 6$). **e** $V_{1/2}$ values from G–V curves analyzed as in **d** with varying $pH_e$ ($pH_e$ 7.0, $n = 6$; $pH_e$ 7.4, $n = 15$; $pH_e$ 8.0, $n = 6$; $pH_e$ 8.5, $n = 7$; $pH_e$ 9.0, $n = 6$; $pH_e$ 9.5, $n = 9$; $pH_e$ 10.0, $n = 9$; $pH_e$ 10.5, $n = 6$). **f** $V_{1/2}$ values

from G–V curves of WT TALK-2 channels activated with 1.0 mM 2-APB ($n = 5$), 5.0 μM oleoyl-CoA ($n = 7$) or inhibited with 1.0 mM TPenA ($n = 8$). Dashed lines in **e**, **f** represent the level of WT (unstimulated) $V_{1/2}$ at $pH_e$ 7.4. **g** Normalized currents from TEVC measurements of oocytes expressing WT ($n = 11$) and mutant L262A ($n = 5$) or L264A TALK-2 channels ($n = 8$), respectively. Channels were activated by increasing $pH_e$ from 5.5 to 10.5 with 0.5 pH increments. Currents were elucidated with a voltage protocol ramped from −120 mV to +45 mV within 3.5 s, analyzed at +40 mV and normalized to pH 10.5. Data shown are the mean ± s.e.m and the number ($n$) of independent experiments is indicated in the figure and supplementary tables 3 and 4.

for pore opening by fitting the G–V curve to a Boltzmann equation. A leftward shift of the G–V curve (without a change in slope) indicates a reduction in the free energy between the closed and open state of the channel. The open state represents the situation of both gates being simultaneously open but voltage can only exert force on the SF gate that functions as voltage sensor. However, when the two gates are energetically coupled then any circumstance that opens the lower gate should also reduce the electrical work to open the SF gate (Figs. 6 and 7). We tested this concept using TPenA that we have shown to hinder lower gate closure. Indeed, the presence of 1 mM TPenA caused a 46.1 ± 4.9 mV leftward shift of the G–V curve, while in TREK-2 $K_{2P}$ channels lacking a lower gate the G–V curve was not affected in the presence of TpenA (Fig. 6a, b, f and Supplementary Table 4). Likewise, 2-APB and LC-CoA that we have shown to open the lower gate also caused a leftward shift of the G–V curve (Fig. 6f and Supplementary Fig. 7a). Further, stabilizing the open state of SF directly by increasing $pH_e$ also shifted the G–V curve leftwards, thus, deprotonation of the pH sensor reduces the free energy difference between the closed and open state of the SF (Fig. 6d, e). This implies that low pH and hyperpolarization induce a similar closed state of the SF. We further tested whether mutations that open the lower gate (L262A and L264A) would affect the extracellular pH sensitivity. Indeed, both mutations shifted the pH-current relationship towards more neutral pH suggesting that opening of the lower gate promoted the deprotonated pH sensor state indicating allosteric coupling of the lower gate to extracellular pH sensor (Fig. 6g and Supplementary Fig. 7b). We further explored this concept of allosteric coupling using the mutations identified in the functional alanine screen of TM2 and TM4. Indeed, all mutations that produced a g-o-f effect also caused a leftward shift of the G–V curve

(Figs. 3a and 7a, b and Supplementary Table 3). Further, the $V_{1/2}$ shift was correlated to the increase in the L145C modification rate (Fig. 7c). Thus, the degree (i.e., frequency) of lower gate opening cause by the g-o-f mutations was correlated to the reduction in electric work to open the SF gate. Accordingly, the largest effect on the G–V curve (i.e., an 84.0 ± 5.3 mV shift) was seen for the L264A mutation that had the largest g-o-f effect, as well as caused the fastest modification of L145C and, thus, the highest $P_O$ of the lower gate (Figs. 3a, k and 7a–c). Actually, this mutation promoted the open state of the lower gate so strongly that 2-APB produced only a minor further current increase (Supplementary Fig. 6d), the speed of L145C MTS-ET modification at +40 mV was not further increased by Rb⁺ (Supplementary Fig. 6e) and TPenA has no marked effect on the G–V curve (Fig. 6c). Therefore, we conclude that the -84 mV shift in $V_{1/2}$ produced by the L264A mutant might roughly represent the mechanical load of the conformational change that the voltage-powered SF gate has to move for opening the lower gate.

## Discussion

The present study on TALK-2 $K_{2P}$ channels provides several lines of evidence suggesting the existence of a lower permeation gate at the cytoplasmic pore entrance in TALK-2. Firstly, access of cysteine modifying reagents to the inner pore cavity is blocked in closed TALK-2 channels, but possible in the presence of activating ligands, suggesting a pore entrance constriction. Furthermore, systematic alanine scanning mutagenesis identified several g-o-f in distal parts of TM2 and TM4 that not only strongly activated TALK-2 but also removed the entrance constriction. Employing TALK-2 homology models based on the TASK-1 and TASK-2 structures revealed that the g-

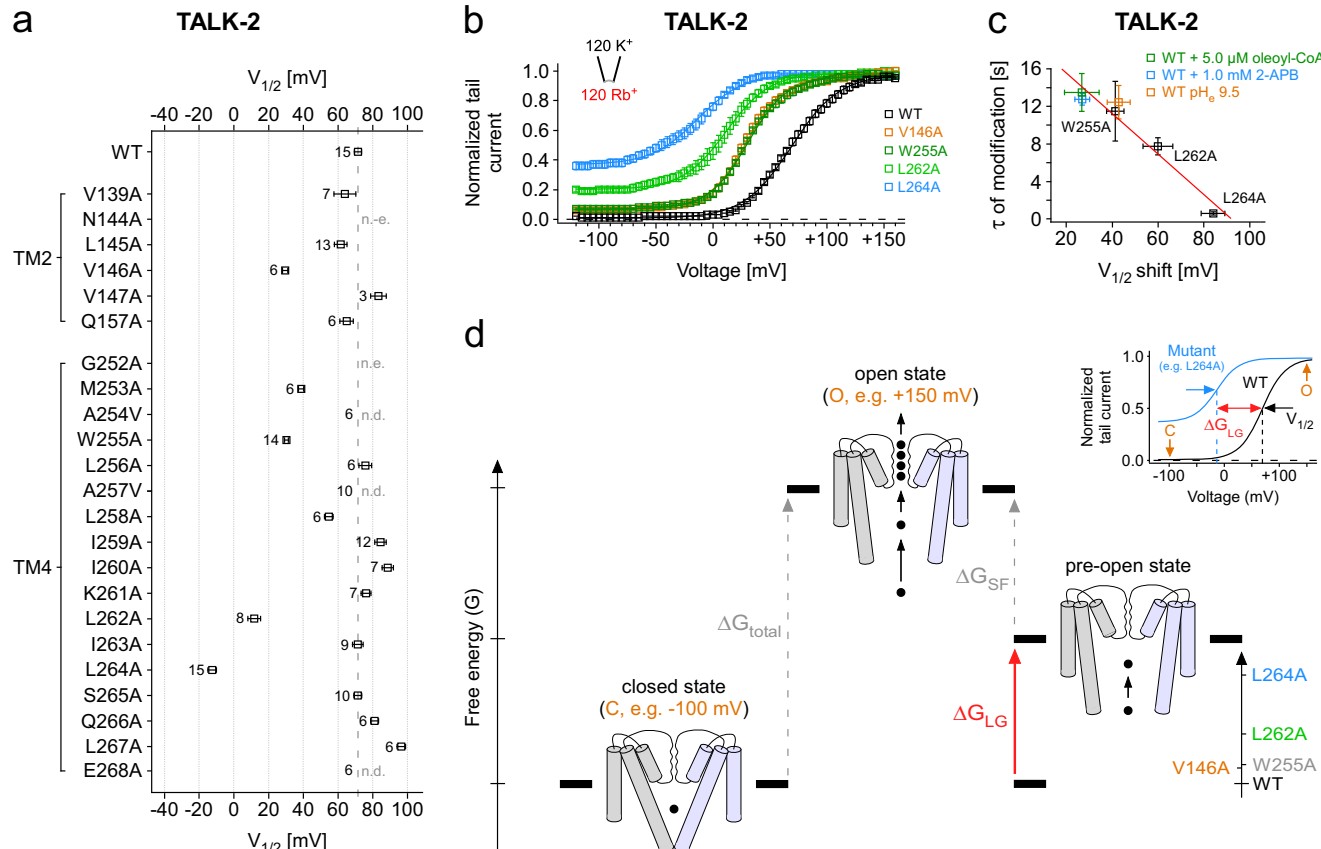

**Fig. 7 | Functional coupling of the SF and the lower gate in TALK-2 K$_{2P}$ channels.**
**a**, **b** G–V curves analyzed from tail currents at -80 mV after 300 ms pre-pulse steps (-120 mV to +160 mV with 5 mV increments) under symmetrical ion conditions with intracellular Rb$^+$ of WT ($n = 15$) and V146A ($n = 6$), W255A ($n = 14$), L262A ($n = 8$), and L264A mutant TALK-2 channels ($n = 15$), respectively (b) and the summary of V$_{1/2}$ values from Boltzmann fits to the corresponding G–V curves (a). Dashed line in a represents the level of WT (unstimulated) V$_{1/2}$ at pH$_e$ 7.4. **c** Correlation of the V$_{1/2}$ shifts of mutant TALK-2 channels at basal and WT TALK-2 channels at indicated conditions with the time constants of modification of TALK-2 L145C channels under the corresponding activatory conditions or in combination with the respective g-o-f

mutation. **d** Simplified energetic scheme depicting the electrical work ($\Delta G = zF\Delta V_{1/2}$) required to open both gates ($\Delta G_{total}$) with the individual contribution of the SF gate ($\Delta G_{SF}$) and lower gate ($\Delta G_{LG}$). Mutations (as indicated in the inlay) that open the lower gate reduced this electrical work as seen in the positive V$_{1/2}$ shifts of the G–V curve. Note, our results actually show that both gates are strongly positively coupled and, thus, the pre-open state (with only the lower gate open) is just a conceptual state to illustrate the energetic contribution of the lower gate. Data shown are the mean ± s.e.m and the number ($n$) of independent experiments is indicated in the figure and supplementary tables 3 and 4. n.d. not determinable, n.e. no expression.

o-f mutations cluster in a region corresponding to the lower gates identified in these channels. Finally, using Rb$^+$ as a probe we show that the identified lower constriction is actually a permeation gate.

## Tight gate coupling defines the gating behavior in TALK-2 channels

In addition to the lower gate, TALK-2 channels are also operated by a SF gate similar to many other K$_{2P}$ channels. We used three assays to dissect the status of the two gates and their coupling in the context of various gating stimuli: (i) the modification rate of L145C reports about the relative $P_O$ of the lower gate, (ii) the G–V curves report on the electrical work required to open the SF filter gate and (iii) the current amplitudes report on the fraction of channels with both gates open (overall channel $P_O$). We found that the three parameters were strongly correlated in all g-o-f mutations suggesting a tight positive coupling of the two gates (Fig. 7d). This concept is also supported by a number of additional findings reported here. Opening the SF by high pH$_e$ or voltage, likewise, opened the lower gate (i.e., increased L145C modification rate). Reciprocally, ligands that opened the lower gate (i.e., increased the rate of L145C modification) such as 2-APB or LC-CoA also positively shifted the G–V curve (opening of the SF gate). Particular striking was the observation that exchange of the permeating ion from

K$^+$ to Rb$^+$ caused a large increase in the L145C modification rate indicating that the mere difference in SF ion occupancy is sufficient to induce a marked structural change at the pore entrance (i.e., opening the lower gate). In structural terms the strong positive gate coupling can be envisioned as rigid connection of the two gates within the protein structure and, thus, opening/closing of one gate also forces the other gate to open/close.

## How is the lower gate coupled to the selectivity filter?

We speculate that movement of TM4 underlies lower gate opening as well as force transduction onto the SF gate to induce its opening in TALK-2. Thus, the TM4 might represent the rigid connection linking the two gates responsible for the tight gate coupling. This hypothesis appears appealing as a similar mechanism has been shown to couple the lower gate to the SF gate in MthK (prokaryotic K$^+$ channel from *Methanobacterium thermoautotrophicum*), based on MD simulations performed with different opening diameters at the lower gate[36]. Here an isoleucine (I84) at the end of TM2 (corresponding to TM4 in TALK-2) was shown to exert force on a threonine (T59) of the SF to promote the conductive SF state. Although members of the TREK/TRAAK sub-family lack a functional lower gate[28,29], the TM4 also moves during temperature-, mechano- or lipid-induced gating (known as the down-

to up-state transition[37,38]) and this movement is also thought to open the SF gate[25,26]. Furthermore, also the lower X-gate in TASK-1 is formed by TM4 and, thus, envisioned to move during gating[31]. Therefore, movement of the TM4 segments coupled to the SF gate might be the general theme in $K_{2P}$ channel gating (as well as other $K^+$ channels). In some channels this movement may result in the formation of a lower gate (e.g., in TASK-1, TASK-2, and TALK-2) and in other channels not (e.g., in TREK-1/-2 and TRAAK), while ultimately the SF is affected in the activation process. How this coupling is realized in atomic detail in TALK-2 warrant future studies and in particular high resolution structural information of different states.

### Physiological relevance of positive gate coupling in TALK-2

The existence of two gates that open and close in a strongly coupled (i.e., concurrently) fashion seams redundant as closure of one gate is sufficient to prevent permeation. However, in TALK-2 channels the two gates have evolved to respond to different stimuli with the SF gate sensing membrane voltage and extracellular pH whereas the lower gate responds to negatively-charged lipids of the cytoplasmic membrane leaflet. Thus, for the individual gate-specific stimuli to become effective strong gate coupling is mandatory as opening of one gate would be functionally silent if the other gate stays closed. Interestingly, this stays in contrast to the multi-sensory gating behavior of the related TREK/TRAAK $K_{2P}$ channel subgroup, where all stimuli (including temperature, voltage, lipids, and pH) are converged directly to the SF representing their only gate.

### Implications of the state-dependent pore access for TALK-2 gating and pharmacology

Functionally, the lower gate in TALK-2 resembles the helix-bundle crossing gate in voltage-gated $K_v$ channels. In $K_v$ channels voltage is thought to open the helix-bundle crossing gate via the voltage-dependent movement of segment 4 (S4)[39,40]. By marked contrast, in $K_{2P}$ channels, voltage opens the SF gate via a voltage-dependent ion binding step[20] that forces the filter in its conductive state. In TALK-2 $K_{2P}$ channels this structural change in the SF appears also to force the lower gate open as apparent in voltage-dependent modification rate of L145C. Upon repolarization, the ion-flux inversion is thought to change the SF ion occupancy leading to its inactivation as visible in the fast reduction of the tail current amplitudes[20]. Intriguingly, in the presence of the pore blocker TPenA the decay of the current is slowed resembling the 'foot in the door' effect seen in $K_v$ channels[33,34]. This suggests that blocker binding to the pore cavity prevented the lower gate from closing. Importantly, this also suggests that TPenA concurrently prevented the SF gate from closing as blocker unbinding from a channel with a non-conductive SF would be electrophysiological invisible and, thus, would not produce the apparent slowing of deactivation. In agreement, in TREK-2 channels (lacking the lower gate) TPenA had no effect on the tail current kinetics indicating that the SF gate can close unhindered with the blocker bound upon inversion of the electric field upon repolarization. What stops now the SF gate from closing with TPenA bound in TALK-2? In $K_v$ channels blocker binding to the pore cavity prevents lower gate closure (the actual structural change is still unknown) suggesting that the pore cavity changes its structure (and possibly its size) in concert with the lower gate closure and this change is hindered (or delayed) by the blocker[41,42]. Thus, we presume that a similar structural change is also occurring in TALK-2 and is also prevented by TPenA. Consequently, because the SF gate and the lower gate are tightly coupled, TPenA also disturbs the SF gate in the closing reaction. In further agreement, we found that TPenA caused a leftward shift of the $G–V$ curve as expected for a blocker that only binds to the pore with both gates open. In single-channel recordings we estimated the basal activity (i.e., the $NP_O$) for WT TALK-2 to about 4 % and, thus, only a small fraction of channels would be sensitive to a pore blocker if only open channels can bind the compound (Fig. 5h). Indeed, we

observed very little current inhibition when TPenA was applied to TALK-2 channels in absence of an activating stimulus, but strong inhibition was observed for activated TALK-2 channels. This behavior was also seen for various other small molecule pore blockers, thus, state-dependent pore inhibition appears to be a defining feature of the TALK-2 channel pharmacology as this was so far not seen in any other $K_{2P}$ channel.

### Gate coupling in other $K^+$ channels: differences and similarities

The established gating cycles of $K_v$ and KcsA (prokaryotic $K^+$ channel from the soil bacterium *Streptomyces lividans*) channels suggest that the activation (lower) gate serves as the primary ion permeation barrier that needs to open to allow current flow, while in a following step the SF inactivates to terminate ion permeation[39,40,43–45]. Thus, in these cases the two gates are coupled sequentially and, in a negative manner[45]. However, it is currently unknown whether the SF is open (conductive) or closed (non-conductive) in the resting (not activated) state of $K_v$ or KcsA channels because it is not possible to directly measure the conductivity of the SF in a closed channel. In TALK-2 $K_{2P}$ channels the SF serves as a gate as well as the voltage sensor. The latter property provides information on stability of the conductive state of the filter that can be extracted from the $V_{1/2}$ value of the corresponding $G–V$ curve. Here, we show that when the lower gate is mostly closed then it requires a large amount of electrical work to open the SF, i.e., a membrane depolarization to $72 \pm 2$ mV ($V_{1/2}$) for half maximal voltage activation. However, when the lower gate is mostly open as seen in the L264A TALK-2 mutant channel this work is strongly reduced ($V_{1/2} = -12 \pm 2$ mV) and, thus, might approximately reflect the work required to open the lower gate. Actually, even very negative potentials cannot close the SF gate to a large degree as seen by the pedestal for the relative $P_O$ that levels off at -0.4 in TALK-2 L264A channels (Fig. 7b). Notably, these results closely resemble the constitutively open phenotype in *Shaker* $K^+$ channels seen with mutations at the assumed hydrophobic seal of the helix-bundle crossing gate. These mutations also result in strong positively shifted $G–V$ curves and high pedestal $P_O$s even at very negative potentials[46]. Interestingly, the two residues with the strongest g-o-f effect (V146A and L264A) in TALK-2 are also hydrophobic and in direct proximity (according to our TASK-1 based homology model) and, thus, might also form a hydrophobic seal[47] that, thereby, could represent the lower gate in TALK-2. Intriguingly, a recent MD simulation study on KcsA suggest that the SF might be non-conductive when the lower (activation) gate is closed as this also implies positive coupling of the two gates in this archetypical $K^+$ channel[47]. Here, using electrophysiological means, we have demonstrated directly that the SF gate in TALK-2 $K_{2P}$ channels opens and closes in concerts with the lower gate. It will be interesting to see whether this concept also applies to other $K^+$ channels possessing two gates such as members of the large family of voltage-gated $K^+$ channels.

## Methods
### Molecular biology

In this study the coding sequences of human $K_{2P}2.1$ TREK-1 (GenBank accession number: NM_172042), human $K_{2P}10.1$ TREK-2 (NM_021161) and human $K_{2P}17.1$ TALK-2 (EU978944.1)/human $K_{2P}17.1$ TASK-4 (NM_031460.3) were used. For $K^+$ channel constructs expressed in oocytes the respective $K^+$ channel subtype coding sequences were subcloned into the oocyte expression vector pSGEM or the dual-purpose vector pFAW which can be used for HEK293 cell expression as well and verified by sequencing. All mutant channels were obtained by site-directed mutagenesis with custom oligonucleotides. Vector DNA was linearized with NheI or MluI and cRNA synthesized in vitro using the SP6 or T7 AmpliCap Max High Yield Message Maker Kit (Cellscript, USA) or HiScribe® T7 ARCA mRNA Kit (New England Biolabs) and stored at −20 °C (for frequent use) and −80 °C (for long term storage).

## Electrophysiological recordings in oocytes

**Two-electrode voltage-clamp (TEVC) measurements.** Electrophysiological studies were performed using the TEVC technique in oocytes. Ovarian lobes were obtained from frogs anesthetized with tricaine. Lobes were treated with collagenase (2 mg/ml, Worthington, type II) in OR2 solution containing (in mM): 82.5 NaCl, 2 KCl, 1 MgCl₂, 5 HEPES (pH 7.4 adjusted with (NaOH/HCl) for 2 h. Isolated oocytes were stored at 18 °C in ND96 recording solution (in mM): 96 NaCl, 2 KCl, 1.8 CaCl₂, 1 MgCl₂, 5 HEPES (pH 7.5 adjusted with NaOH/HCl) supplemented with Na-pyruvate (275 mg/l), theophylline (90 mg/l), and gentamicin (50 mg/l). Oocytes were injected with 50 nl of cRNA for WT or mutant TALK-2 and incubated for 2 days at 18 °C. Standard TEVC measurements were performed at room temperature (21 - 22 °C) with an Axoclamp 900 A amplifier, Digidata 1440 A, and pClamp10 software (Axon Instruments, Molecular Devices, LLC, USA). Microelectrodes were fabricated from glass pipettes, back-filled with 3 M KCl, and had a resistance of 0.2–1.0 MΩ.

**Inside-out patch-clamp measurements.** Oocytes were surgically removed from anesthetized adult females, treated with type II collagenase (Sigma-Aldrich/Merck, Germany) and manually defolliculated. 50 nl of a solution containing the K⁺ channel specific cRNA was injected into Dumont stage V - VI oocytes and subsequently incubated at 17 °C in a solution containing (mM): 54 NaCl, 30 KCl, 2.4 NaHCO₃, 0.82 MgSO₄ x 7 H₂O, 0.41 CaCl₂, 0.33 Ca(NO₃)₂ x 4 H₂O and 7.5 TRIS (pH 7.4 adjusted with NaOH/HCl) for 1–7 days before use. Electrophysiological recordings: Excised patch measurements in inside-out configuration under voltage-clamp conditions were performed at room temperature (22 – 24 °C). Patch pipettes were made from thick-walled borosilicate glass GB 200TF-8P (Science Products, Germany), had resistances of 0.2–0.5 MΩ (tip diameter of 10 - 25 μm) and filled with a pipette solution (in mM): 120 KCl, 10 HEPES and 3.6 CaCl₂ (pH 7.0 - 8.5 adjusted with KOH/HCl) or 120 KCl, 10 AMPSO and 3.6 CaCl₂ (pH 9.0 - 10.5 adjusted with KOH/HCl). Intracellular bath solutions and compounds were applied to the cytoplasmic side of excised patches for the various K⁺ channels via a gravity flow multi-barrel pipette system. Intracellular solution had the following composition (in mM): 120 KCl, 10 HEPES, 2 EGTA and 1 Pyrophosphate (pH adjusted with KOH/HCl). In other intracellular bath solutions, K⁺ was replaced by Rb⁺ (pH 7.4 adjusted with RbOH/HCl). Currents were recorded with an EPC10 amplifier (HEKA electronics, Germany) and sampled at 10 kHz or higher and filtered with 3 kHz (-3 dB) or higher as appropriate for sampling rate.

**Inside-out single channel patch-clamp measurements.** Single channel patch-clamp measurements in the inside-out configuration were performed under voltage-clamp conditions with oocytes. Briefly, the vitelline membranes of the oocytes were manually removed after shrinkage by adding mannitol to the bath solution. All experiments were conducted at room temperature (21 - 22 °C) 1–2 days after injection of 50 nl TALK-2 cRNA. Borosilicate glass capillaries GB 150TF-8P (Science Products, Germany) were pulled with a DMZ-Universal Puller (Zeitz Instruments, Germany) and had a resistance of 4–6 MΩ when filled with pipette solution containing (in mM): 120 KCl, 10 HEPES, 3.6 CaCl₂ (pH 8.5 adjusted with KOH/HCl). The bath solution had the following composition (in mM): 120 KCl, 10 HEPES, 2 EGTA, and 1 Pyrophosphate (pH 8.5 adjusted with KOH/HCl). Gap-free voltage pulses of −100 mV were continuously applied. Single channel currents were recorded with an Axopatch 200B amplifier, a Digidata 1550B A/D converter and pClamp10 software (Axon Instruments, Molecular Devices, LLC, USA) and were sampled at 15 kHz with the analog filter set to 5 kHz. Additionally, data was digitally filtered by 2 kHz (Lowpass, Bessel) with ClampFit10 before analysis. Data was analyzed with ClampFit10 and Origin 2016 (OriginLab Corporation, USA). The single channel search tool of the ClampFit10 software was used to identify and analyze the single channel events in a time frame of 60 s. A group of single channel events (minimum 3 events) has been classified as a burst, if the time between two single channel openings were lower than five times of the short-closed time (5 x $t_{C1}$).

## Animals

The investigation conforms to the guide for the Care and Use of laboratory Animals (NIH Publication 85-23). For this study, sixty female *Xenopus laevis* animals were used to isolate oocytes. Experiments using *Xenopus* toads were approved by the local ethics commission.

## Drugs, chemical compounds, and bioactive lipids

2-aminoethoxydiphenyl borate (2-APB) (Sigma-Aldrich/Merck, Germany), BL-1249, A1899 (Tocris Bioscience, Germany), AVE0118, S9947 (Axon Medchem, Germany) and oleoyl-CoA (LC-CoA 18:1) (Avanti Polar Lipids, USA) were prepared as stocks (1 - 100 mM) in DMSO, stored at −80 °C and diluted to the final concentration in the intracellular recording solution. Tetra-pentyl-ammonium chloride (TPenA) (Sigma-Aldrich/Merck, Germany) and (2-(Trimethylammonium)ethyl) MethaneThioSulfonate Chloride (MTS-ET) (Toronto Research Chemicals, USA) were directly dissolved to the desired concentration in the intracellular recording solution prior to each experiment. MTS-ET was used immediately after dilution for maximally 5 min.

## Homology modeling - TALK-2 models based on TASK-1 or TASK-2

Human TASK-1 crystallographic structure (PDB ID: 6RV3, chains A, B)[31] and mouse TASK-2 cryo-electron microscopy (Cryo-EM) structure (PDB ID: 6WLV, chains A, B)[30] were used as templates to build the TALK-2 models. First, the structures were prepared using 'Protein preparation wizard' module of the Schrödinger's suite software (Protein Preparation Wizard; Epik, Schrödinger, LLC, New York, NY, 2018-4; Impact, Schrödinger, LLC, New York, NY, 2018-4; Prime, Schrödinger, LLC, New York, NY, 2018-4). Charges and parameters were assigned according to the force field OPLS-2005[48]. The missing residues in TASK-1 (149 - 151 from chain A and 150 - 151 from chain B) were modeled using 'crosslink protein' tool from the Schrödinger suite. The sequence of TALK-2 (KCNK17) protein was obtained from the GenBank database[49]. Alignments between the template protein (TASK-1 or TASK-2) sequences and the TALK-2 sequence were performed using the Smith-Waterman algorithm[50] with BLOSUM62[51] matrix for scoring the alignment. The modeling was carried out using the BioLuminate[52] module of the Schrödinger suite. The standard modeling protocol was used, which consists of a search for rotamers for non-conserved residues and loops to eliminate clashes and then an energetic minimization with the OPLS2005 force field in vacuum. The protonation states of the TALK-2 models were predicted at a pH of 7.0 with PROPKA[53], and another energetic minimization of the hydrogen atoms was performed using the conjugate gradient method[54] implemented in the Schrödinger suite. The sequence of the TALK-2 model based on 6RV3 extends from residue R15 to K282 and based on 6WLV from residue T22 to H278, respectively. It was verified with the procheck[55] software that 91.2 % and 93.5 % of the residues are in the most favored regions for the models based on 6RV3 and 6WLV, respectively.

Alignments:

6RV3: MKRQNVRTLALIVCTFTYLLVGAAVFDALESEPELIERQRLELR QQELRARYN - LSQGGYEELERVVLRLKPHKAGVQ - - - - - - - - - WRFAGSF YFAITVITTIGYGHAAPSTDGGKVFCMFYALLGIPLTLVMFQSLGERINTLV RYLLHRAKKGLGMRRADVSMANMVLIGFFSCISTLCIGAAAFSHYEHWTF FQAYYYCFITLTTIGFGDYVALQKDQALQTQPQYVAFSFVYILTGLTVIGAF LNLVLRFMTMNAEDEKRDAENL

TALK-2 (TASK-4): RGCAVPSTVLLLLAYLAYLALGTGVFWTLEGRAA QDSSRSFQRDKWELLQNFTCLDRPALDSLIRDVVQAYKNGASLLSNTTSM GRWELVGSFFFSVSTITTIGYGNLSPNTMAARLFCIFFALVGIPLNLVVLNRL GHLMQQGVNHWASRLGGTWQDPDKARWLAGSGAL - LSGLLLFLLLPPL

LFSHMEGWSYTEGFYFAFITLSTVGFGDYVIG ~ MNPSQRYPLWYKNMVSL
WILFGMAWLALIIKLILSQLETPGRVCSCCHHSSK

6WLV: GPLLTSAIIFYLAIGAAIFEVLEEPHWKEAKKNYYTQKLHLLKEF
PCLSQEGLDKILQVVSDAADQGVAITGNQT ~ FNNWNWPNAMIFAATVIT
TIGYGNVAPKTPAGRLFCVFYGLFGVPLCLTWISALGKFFGGRAKRLGQFL
TRRGVSLRKAQITCTAIFIVWGVLVHLVIPPFVFMVTEEWNYIEGLYYSFITI
STIGFGDFVAGVNPSANYHALYRYFVELWIYLGLAWLSLFVNWKVSMFVE
VHKAIKKRR

TALK-2 (TASK-4): TVLLLLAYLAYLALGTGVFWTLEGRAAQDSSRSF
QRDKWELLQNFTCLDRPALDSLIRDVVQAYKNGASLLSNTTSMGRWELV
GSFFFSVSTITTIGYGNLSPNTMAARLFCIFFALVGIPLNLVVLNRLGHLMQ
QGVNHWASRLGGTWQDPDKARWLAGSGALLSGLLLFLLLPPLLFSHME
GWSYTEGFYFAFITLSTVGFGDYVIGMNPSQRYPLWYKNMVSLWILFGM
AWLALIIKLILSQLETPGRVCSCCH

Cysteine mutations were introduced via PyMOL after the respective model was built.

## Data acquisition and statistical analysis

Data analysis and statistics were done using Fitmaster (HEKA electronics, version: v2x73.5, Germany), Microsoft Excel 2021 (Microsoft Corporation, USA), and Igor Pro 9 software (WaveMetrics Inc., USA). Recorded currents were analyzed from stable membrane patches at a voltage defined in the respective figure legend or with a voltage protocol as indicated in the respective figure. The fold activation (fold change in (tail) current amplitude) of a ligand (drug or bioactive lipid) was calculated from Eq. 1:

$$\text{Fold activation(FA)} = \frac{I_{\text{activated}}}{I_{\text{basal}}} \tag{1}$$

where $I_{\text{activated}}$ represents the stable current level in the presence of a given concentration of a respective ligand and $I_{\text{basal}}$ the measured current before ligand application. Percentage inhibition or residual currents upon blocker application for a ligand (drug or bioactive lipid) was calculated from stable currents of excised membrane patches using the following Eqs. 2 and 3:

$$\%\text{inhibition} = \left(1 - \left(\frac{I_{\text{inhibited}}}{I_{\text{basal}}}\right)\right) * 100 \tag{2}$$

$$\text{Residual current} = 100 - \%\,\text{inhibition} \tag{3}$$

where $I_{\text{inhibited}}$ refers to the stable current level recorded in the presence of a given concentration of a drug or bioactive lipid and $I_{\text{basal}}$ to the measured current before ligand application. The macroscopic half-maximal concentration-inhibition relationship of a ligand was obtained using a Hill-fit for dose-response curves as depicted in Eq. 4:

$$\%\,\text{inhibition/activation} = \frac{I_{base} + (I_{\max} - I_{base})}{\left\{1 + \left[\frac{x_{1/2}}{x}\right]^{rate}\right\}} \tag{4}$$

where base and max are the currents in the absence and presence of a respective ligand, x is the concentration of the ligand, $x_{1/2}$ is the ligand concentration at which the activatory or inhibitory effect is half-maximal, rate is the Hill coefficient. For analysis of activation and deactivation time constants ($\tau$) as well as the time constants of MTS modification ($\tau$) current traces were fitted with a mono-exponential equation as depicted in 5:

$$y_0 + A^{\left\{\frac{-(x-x_0)}{\tau}\right\}} \tag{5}$$

Conductance-voltage ($G-V$) relationships were determined from tail currents recorded at a holding potential ($V_H$) of −80 mV after 300 ms depolarizing steps as indicated in the respective figure legend.

Data were analyzed with a single Boltzmann fit following Eq. 6:

$$G(V) = \frac{I_{\max}}{1 + e^{\frac{(V - V_{1/2})}{slope}}} + I_{base} \tag{6}$$

where $V_{1/2}$ represents the voltage of half-maximal activation, s is the slope factor and $I_{\max}$ and $I_{\text{base}}$ represent the upper and lower asymptotes. Under appropriate experimental conditions, you can use slope to calculate the valence (charge) of the ion moving across the channel. Slope equals R * T / z * F where R is the universal gas constant, $T$ is temperature in K, $F$ is the Faraday constant, and $z$ is the valence.

Throughout the manuscript all values are represented as mean ± s.e.m. with $n$ indicating the number of individual executed recordings of single patches or oocytes. Data from independent measurements (biological replicates) were normalized and fitted independently to facilitate averaging. Statistical significance between two groups (respective datasets) was validated using an unpaired, two-sided $t$-test. The results of statistical analyses are presented as follows: $*P \le 0.05$, $**P \le 0.01$ and $***P \le 0.001$. The exact $P$ values are indicated in the figures. Zero current levels were indicated using dotted lines in all figures. Image processing and figure design was done using Igor Pro 9 (64 bit) (WaveMetrics, Inc., USA), PyMOL 2.4.1 (Schrödinger, LLC), and Canvas X Draw (Version 20 Build 544) (ACD Systems, Canada).

## Reporting summary

Further information on research design is available in the Nature Portfolio Reporting Summary linked to this article.

## Data availability

Data supporting the findings of this manuscript are available from the corresponding authors upon request. The source data underlying Figs. 1g, h, 2c, 3a, 3e–h, 3k, 4e, 4i, 5e–g, 6a–g, and 7a–c, and Supplementary Figs. 1c, e, g, 4a, 5a, 5d, 6c, d, and 7 are provided as Source data file. The accession codes for the PDB structures used to build the models are PDB ID: 6RV3 and 6WLV. Source data are provided with this paper.

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

## Acknowledgements

We thank the members of our laboratories for their technical support and helpful comments on the manuscript. FONDECYT−ANID (grant number 3210774) supported M.B. The Deutsche Forschungsgemeinschaft (DFG, German Research Foundation) supported N.D. (DE1482-9/1) and T.B. (BA 1793/6-2)/ M.S. (SCHE 2112/1-2) as part of the Research Unit FOR2518, *DynIon*.

## Author contributions

N.D., T.B., and M.S. conceived and supervised the project; L.C.N., E.B.R., B.C.J., J.L., B.E., and M.S. performed patch-clamp experiments and analyzed the data. S.R. and N.D. planned and generated mutant constructs for two-electrode voltage-clamp experiments. F.-R.S. and S.R. performed, and A.K.K. and S.R. analyzed TEVC experiments. N.D. planned, S.R. performed, and A.K.K. analyzed single-channel recordings. S.C. planned mutant constructs for patch-clamp experiments. M. B. generated the TALK-2 homology models. M.S. prepared figures; T.B. and M.S. wrote the article with contributions from all authors.

## Funding

## Competing interests

The authors declare no competing interests.
