## [Peer Review File · Nature Communications]

Ion occupancy of the selectivity filter controls opening of a cytoplasmic gate in the K2P channel TALK-2REVIEWER COMMENTS

Reviewer #1 (Remarks to the Author):

Two-pore domain potassium (K2P) channels are typically regarded as potassium leak channels with minimal voltage gating. However, the intricate regulation of K2P channels is critical for understanding their complex gating mechanisms. Common consensus suggests that K2P channels are primarily gated through their selectivity filter (SF). Nonetheless, the crystalline structure and predictive modeling of certain K2P members (TASK1, TASK2, TALK2) indicate an additional gating mechanism, known as the lower gate, situated at the cytoplasmic end of the channel, extending the pore pathway. Neelsen and colleagues present empirical evidence of this lower gate in the TALK2 channel, utilizing cysteine mutagenesis techniques and pharmacological tools. Their research reveals a pronounced bidirectional coupling between the SF and lower gates, where the activation of one gate induces the opening of the other. This insight significantly enhances our comprehension of K2P channel gating mechanisms, especially those featuring a lower gate. Moreover, this concept of robust, reciprocal coupling might also be applicable to other potassium channels with similar gating structures, aiding in the broader understanding of their operational dynamics.

The study employs a range of electrophysiological methods, providing compelling evidence supporting the proposed model of the coupled lower and SF gates. The experimental approach is innovative and meticulously executed, underpinning the study's conclusions through both textual and diagrammatic representations. While the paper primarily investigates K2P channels, its implications extend to other ion channel families, making it a valuable resource for a broad scientific audience due to its clear and comprehensive presentation.

Despite the well-executed experiments and extensive controls, certain aspects of the study could benefit from additional clarification to enhance and solidify the manuscript's findings.

Major scientific comments

Figure 1 :

To probe the cytoplasmic constriction, the authors showed that MTS-ET did not affect the basal current carried by TASK-L145C whereas it induced an inhibition on the pre-activated channel by 2-ABP. They concluded that, when the channel is in its low-activity state, the MTS-ET has no access to the Cysteine. But the absence of effect does not necessarily mean that the MTS-ET did not bind. As the channel activity is really low, it might be difficult to observe an inhibition. The author should add an additional experiment in which they apply first MTS-ET and then apply 2-ABP. If the MTS-ET did not prevent the activating effect of 2-ABP, this would demonstrate that the MTS-ET is not covalently linked to the Cys allowing to conclude that the access was blocked.

Supplementary figure 1

Sup Figure 1a: Authors test the impact of the application of MTS-ET on TALK2-WT. To make sure MTS-ET does not have any effect on the channel, MTS-ET should also be applied in the presence of BL-1249. Besides, BL-1249 should also be applied before the application of MTS-ET to compare the current fold increase induced by BL-1249 before and after MTS-ET application. This way, authors will be sure MTS-ET does not have any residual effect on activated channels after MTS-ET application and wash-out.

Authors should carry out the same experiment as in Figure 1a using BL-1249 on TALK2 L145C channels to provide additional proof of similar activity between TALK2 WT and TALK2 L145C.

Sup Figure 1c: Authors conclude that MTS-ET does not have any effect on pharmacologically activated TALK2-WT current. Statistical comparison of the currents once the steady state is reached should be carried out on several independent recordings. Furthermore, on the representative trace, the steady state in the presence of 2-APB (before MTS-ET application) is not fully reached. In the case of Oleoyl-

Coa, the current is unstable, making the conclusion difficult on the effect of MTS-ET (that has an immediate effect on the current by the way).

Figure 2

The experiment uses Rb⁺ as a read-out of the open state of the lower gate: (l.23, p.7) "Rb⁺ needs to pass the lower constriction". Please justify this assertion more thoroughly to make the experimental paradigm stronger.

Figure 2c: To provide a complete quantification of the relation between the Rb⁺ activation potential and the 2-APB concentration, authors should add points at higher 2-APB concentrations. Authors should also comment on the 2-APB concentration for which the Rb⁺ activation is maximal regarding Figure 1h (i.e. the minimal 2-APB concentration needed for a maximal activation, as well as for the fastest effect of MTS-ET).

Figure 3

Authors used the L264A mutant, more than others, to carry out a more thorough analysis of the Po, close, and open times of the channel. Could the authors justify this choice in the text?

Figure 3k: are the MTS-ET modification rates statistically different between mutants? Please provide the results of the statistical test.

Figure 5

Figure 5b: please justify why TREK2 is used as a control and not TREK1 as it was in figure 1g.

Figures 5a,b: Are the V_{1/2} values with TPenA compared to that in the absence of TPenA significantly different?

Figures 5f,g: Results of statistical tests (that are supposed to be significant) should be added to enhance the results.

To further validate the model, it would be interesting to experiment on the tail current in the presence or absence of TPenA on 2-APB-activated TALK2. There should not be any tail current cross-over, as in TREK2.

Figure 6

Figure 6b: Same remark as in Figure 5b: why is TREK2 used as a control and not TREK1 as in Figure 1g? or Why TREK1 is used in Fig1g?

Figure 6f: It would be valuable to add the V_{1/2} in the presence of TPenA in the graph and comment on the similarity of the values.

Figure 6d: The authors should use a different set of colors for different pHs compared to the presence of TPenA for a matter of clarity.

Minor comments

Figure 2b: the representative trace shown is not fully representative of the results reported in Figure 2c. No current increase is observed at 0.5mM 2-ABP for TREK1.

Please provide the precise concentrations of the different compounds used in the article, in particular the 2-APB, Oleoyl-Coa working concentrations (on traces).

May the authors develop in the discussion the following points?

Overall, the results of statistical tests should be systematically mentioned in the figures (including non-significant results). The authors show a strong reciprocal coupling between the lower gate and the SF gate. What additional properties does this coupling provide concerning the gating mechanism of the channel compared to a K2P channel without the lower gate? The authors point out that it has an impact on the pharmacology of the channel. But physiologically, are there some situations where the lower gate plays a role in the regulation of the channel?

In several experiments, Rb⁺ is used to potentiate TALK2 currents through its supposed action on the SF. However, considering the strong reciprocal coupling between the lower gate and the SF, how can Rb⁺ have a further activation effect on TALK2 in conditions where the lower gate is opened (in presence of 2-APB for example – figure 2)?

In the method section

- p.23, l.2: "Statistical significance between two groups (respective datasets) was validated using an unpaired Student's t-test or Wilcoxon rank test after f-test application."

The Wilcoxon rank test should be used if data are not normally distributed and do not depend on f-test results. The latter enables to determine if the variance between the 2 groups is equal. If this is the case, a Student's t-test should be carried out, if not, a Welch's t-test needs to be used.

- p.4, l.15: "Furthermore, thus TALK-2 channels are highly and specifically expressed in the human pancreas and are considered as a risk factor for the pathogenesis of type 2 diabetes." Considering the meaning of the sentence, "thus" should be removed.

- p.5, l.10: "blocker" → "blockers"

- p.5, l.16: "Further, we demonstrate that the lower gate produced a state-dependent blocker pharmacology that is unique in K2P channels."

Please rephrase the sentence. Maybe "blocker" should simply be removed?

- p.6, l.8: "the mechanisms how they open the ion permeation pathway are currently unknown."

Please rephrase the sentence. (maybe replace "how" by "through which"?)

- p.7, l.6: "result" → "results"

- p.8, l.29: "spend" → "spent"

- p.8, l.31: "are" → "is"

- p.13, l.9: "Employing a TALK-2 homology models based on the TASK-1 and TASK-2 structures." "a" should be removed.

- p.15, l.2: "suggest" → "suggests"

- p.15, l.3: "suggest" → "suggests"

- p.15, l.12: "possible" → "possibly"

- p.15, l.25: "blocker" → "blockers"

- p.28, l.21: "similar" → "similarly"

- p.21, l.22: "with" → "where"

- p.22, l.6, l.22: "with" → "where"

Reviewer #3 (Remarks to the Author):

In this manuscript, Neelsen and colleagues show that the K2P potassium channel TALK-2 contains a cytoplasmic gate, and that this gate and the gate formed by the selectivity filter are positively coupled. The study is extremely well conducted, and the data fully support the conclusion. This is clearly original work but one wonders if the originality is sufficient for it to be published in Nature Communications. The presence of two gates is classic in channels of the different potassium channels subfamilies, with an upper gate formed by the selectivity filter and a lower gate coupled to the cytoplasmic tail in K2P channels or in the bacterial KcsA channel, or to a voltage-sensing membrane domain for voltage-gated potassium channels. Furthermore, these two gates have been shown to be

positively coupled in the K2P channel KCNK0 (reference missing in the manuscript: Ben-Abu et al, Nature Structural & Molecular Biology 16, 71–79, 2009) and in KcsA (Heer et al, eLife 6, 2017), and negatively coupled in voltage-gated potassium channels (reference missing in the manuscript: Panyi & Deutsch, J Gen Physiol 129,403-418, 2007). It has been proposed by Ben-Abu and colleagues that this inverse coupling in leak K2P channels and voltage-gated potassium serve electrical signaling, as leak channels have evolved to be predominantly open, whereas voltage-gated channels must inactivate once activated by a depolarization during an action potential.

Specific comments: Ben-Abu's study should be cited and discussed in the manuscript. Point 4 of the synopsis and the abstract should be corrected because studies on voltage-gated potassium channels show a negative coupling between the two gates. The manuscript would benefit from a discussion of the physiological significance of the TALK-2 gating (which signals are integrated and what is the impact of positive-coupling between the two gates).

NCOMMS-23-60305 RESPONSE TO REVIEWER COMMENTS

We thank the reviewers for their positive and helpful comments in evaluating this manuscript. We suggest we can address all these comments as indicated below and hope that the improved manuscript is now suitable for publication.

Reviewer #1 (Remarks to the Author):

Despite the well-executed experiments and extensive controls, certain aspects of the study could benefit from additional clarification to enhance and solidify the manuscript's findings.

Major scientific comments

Figure 1 :

To probe the cytoplasmic constriction, the authors showed that MTS-ET did not affect the basal current carried by TASK-L145C whereas it induced an inhibition on the pre-activated channel by 2-ABP. They concluded that, when the channel is in its low-activity state, the MTS-ET has no access to the Cysteine. But the absence of effect does not necessarily mean that the MTS-ET did not bind. As the channel activity is really low, it might be difficult to observe an inhibition. The author should add an additional experiment in which they apply first MTS-ET and then apply 2-ABP. If the MTS-ET did not prevent the activating effect of 2-ABP, this would demonstrate that the MTS-ET is not covalently linked to the Cys allowing to conclude that the access was blocked.

We thank the reviewer for this suggestion. We have performed several individual experiments (n = 6) where we first applied MTS-ET and then checked for the degree of 2-APB activation. Indeed, TALK-2 L145C channels can be robustly activated with 1 mM 2-APB after MTS-ET application to the low-activity state, indicating that the modifying probe is not bound to the cysteine residue within the cavity because the access was blocked. We now include such an example trace in Fig. 1d.

Supplementary figure 1

Sup Figure 1a: Authors test the impact of the application of MTS-ET on TALK2-WT. To make sure MTS-ET does not have any effect on the channel, MTS-ET should also be applied in the presence of BL-1249. Besides, BL-1249 should also be applied before the application of MTS-ET to compare the current fold increase induced by BL-1249 before and after MTS-ET application. This way, authors will be sure MTS-ET does not have any residual effect on activated channels after MTS-ET application and wash-out. Authors should carry out the same experiment as in Figure 1a using BL-1249 on TALK2 L145C channels to provide additional proof of similar activity between TALK2 WT and TALK2 L145C.

Following the reviewer's suggestion we now include (and replace) results in Supplementary Fig. S1 indicating that MTS-ET application neither has an effect on BL-1249-activated TALK-2 WT channels nor alters the fold activation with BL-1249 in two repetitive applications, i.e., when applied in the low-activity state between them. Further, we quantified the fold activation with 50 μ M BL-1249 for WT

TALK-2 channels and mutants used for MTS modification. Strikingly WT, L145C and Q266C TALK-2 channels show similar activity in the presence of the activator. Finally, we would like to emphasize that BL-1249 solely was used to demonstrate the expression of the respective channels and so to validate the lack of MTS effect in the low-activity state. Please also note that the main focus in the manuscript is on the small molecule compound 2-APB and the natural ligand oleoyl-CoA.

Sup Figure 1c: Authors conclude that MTS-ET does not have any effect on pharmacologically activated TALK2-WT current. Statistical comparison of the currents once the steady state is reached should be carried out on several independent recordings. Furthermore, on the representative trace, the steady state in the presence of 2-APB (before MTSET application) is not fully reached. In the case of Oleoyl-Coa, the current is unstable, making the conclusion difficult on the effect of MTS-ET (that has an immediate effect on the current by the way).

We agree with the reviewer that the control experiments are not ideal; however, we respectfully disagree that this makes the conclusion difficult. The experiments regarding WT TALK-2 channels in the presence of either 2-APB or oleoyl-CoA are intended to demonstrate that, upon application of MTS-ET for a specific time duration, no dramatic changes occur, as seen in the case of channels with an introduced and modifiable cysteine residue, e.g., TALK-2 L145C activated with 2-APB or oleoyl-CoA. In the experiment with 2-APB, saturation is even noticeable, albeit distorted solely by scaling of the time axis. In the case of TALK-2 channels activated by oleoyl-CoA, a slight run-down is observed (that indeed can occur), which remains unchanged upon the application of MTS-ET. In conclusion, we selected these two experiments as examples because they also illustrate the washout of the two activators after MTS-ET application, thus reflecting the retained functionality of the TALK-2 channels.

Figure 2

The experiment uses Rb⁺ as a read-out of the open state of the lower gate: (l.23, p.7) “Rb⁺ needs to pass the lower constriction”. Please justify this assertion more thoroughly to make the experimental paradigm stronger.

In our opinion, there is no alternative way that Rb⁺ activates TALK-2 channels then via direct interaction with the selectivity filter. Therefore, Rb⁺ has to pass the lower gate to reach the filter when the ion is applied from the cytosolic side.

Figure 2c: To provide a complete quantification of the relation between the Rb⁺ activation potential and the 2-APB concentration, authors should add points at higher 2-APB concentrations. Authors should also comment on the 2-APB concentration for which the Rb⁺ activation is maximal regarding Figure 1h (i.e. the minimal 2-APB concentration needed for a maximal activation, as well as for the fastest effect of MTS-ET).

We agree with the reviewer’s comment that quantification of the activatory potential of Rb⁺, especially for TALK-2 channels, could be improved including higher 2-APB concentrations. Unfortunately, in our

hands, 2-APB is solved as 100 mM stocks in DMSO and as such with 2 mM final 2-APB concentration in the recording solution DMSO reaches the maximal level of 2 % that could be used without producing side effects.

With regards to the second query about the 2-APB concentration needed to achieve maximal channel activation, additive Rb^+ activation and fastest MTS-ET modification, we have stated in the main text that 2 mM 2-APB causes the maximal current activation in K^+ and the fastest MTS-ET modification rate (p7., line 7-12) for the L145C mutant TALK-2 channel. In contrast, the Rb^+ -induced current for WT TALK-2 channels investigated in Fig. 2 is bigger with 2 mM than with 1 mM 2-APB resulting from the slightly reduced apparent affinity of the WT for the activator 2-APB as shown in Fig. 1g.

Figure 3

Authors used the L264A mutant, more than others, to carry out a more thorough analysis of the Po, close, and open times of the channel. Could the authors justify this choice in the text?

The choice of the TALK-2 L264A mutant for further analysis, i.e., single-channel recordings are based on the biggest g-o-f effect that was identified in the systematic alanine screening approach. This point is now mentioned in the main text.

Figure 3k: are the MTS-ET modification rates statistically different between mutants? Please provide the results of the statistical test.

The MTS-ET modification rates are statistically different for the respective g-o-f mutations in combination with L145C. Only the direct comparison between W255A and L262A shows no significant difference. The results are highlighted in the figure and are mentioned in the figure legend.

Figure 5

Figure 5b: please justify why TREK2 is used as a control and not TREK1 as it was in figure 1g.

There is no compelling reason for preferring to study one channel over the other in both sections. The 2-APB sensitivity is nearly identical as well the inhibition of TPenA. The crucial characteristic that the reference channel must possess is that, unlike TALK-2, it lacks an inner gate at the cytoplasmic pore entrance (e.g., TREK-1, TREK-2 or TRAAK).

Figures 5a,b: Are the $V_{1/2}$ values with TPenA compared to that in the absence of TPenA significantly different?

We assume that reviewer #1 refers to Figure 6a, b here? These experiments are shown in Fig. 6a, b. The $V_{1/2}$ values in the absence and presence of TPenA are significantly different for TALK-2 WT in Fig. 5a and not different for TREK-2 WT in Fig. 5b. Besides, either TREK-1, TREK-2 or TRAAK could have been used as a comparison here in Fig. 5b again.

Figures 5f,g: Results of statistical tests (that are supposed to be significant) should be added to enhance the results.

The results of the statistical tests are now included in the figure and in the figure legend.

To further validate the model, it would be interesting to experiment on the tail current in the presence or absence of TPenA on 2-APB-activated TALK2. There should not be any tail current cross-over, as in TREK2.

We thank the reviewer for this valid suggestion and tested the effect of TPenA on 2-APB-activated TALK-2 WT channels. Strikingly, we found no tail current cross-over with activated TALK-2 WT channels, as we would expect if 2-APB primarily opened the lower gate. The new result is now included in Supplementary Fig. 6a and reads as follows in the text:

Further, TPenA inhibition had likewise little effect on tail current kinetics of 2-APB-activated TALK-2 channels indicating that 2-APB preferentially opens the lower gate in TALK-2 (Supplementary Fig. 6a).

Figure 6

Figure 6b: Same remark as in Figure 5b: why is TREK2 used as a control and not TREK1 as in Figure 1g? or Why TREK1 is used in Fig1g?

Please find the explanation before.

Figure 6f: It would be valuable to add the $V_{1/2}$ in the presence of TPenA in the graph and comment on the similarity of the values.

We added the data point for the $V_{1/2}$ in the presence of TPenA in the graph and mention the similarity to the 2-APB or oleoyl-CoA activation in the main text.

Figure 6d: The authors should use a different set of colors for different pHs compared to the presence of TPenA for a matter of clarity.

We fully agree with this comment and have changed the color set for the G-V curves at different pHs.

Minor comments

Figure 2b: the representative trace shown is not fully representative of the results reported in Figure 2c. No current increase is observed at 0.5mM 2-ABP for TREK1.

We agree with the reviewer that, at the first glance, the sample measurement of TREK-1 may seem unrepresentative. We have therefore replaced the trace with a more suitable one.

Please provide the precise concentrations of the different compounds used in the article, in particular the 2-APB, Oleoyl-Coa working concentrations (on traces).

We have now included the precise compound concentrations used for individual recordings throughout the figures.

May the authors develop in the discussion the following points?

Overall, the results of statistical tests should be systematically mentioned in the figures (including non-significant results).

We have now included the results of the statistical tests in all figures, where applicable.

The authors show a strong reciprocal coupling between the lower gate and the SF gate. What additional properties does this coupling provide concerning the gating mechanism of the channel compared to a K2P channel without the lower gate?

We thank the reviewer for this question. The reciprocal coupling of the two gates illustrates, on the one hand, that even subtle changes in the filter can lead to structural alterations in the distant cytoplasmic part (lower gate) of the channel. Essentially, the presence of a second gate and its coupling to the filter gate influence the pharmacology of the channels as demonstrated for the inhibitors. However, we think of greater importance is the understanding of the physiological significance of reciprocal gate coupling in TALK-2 channels, as rightly pointed out by the reviewer in the subsequent question. We discuss this point further in context of the next question.

The authors point out that it has an impact on the pharmacology of the channel. But physiologically, are there some situations where the lower gate plays a role in the regulation of the channel?

We thank the reviewer for the question. One could ask what the physiological significance of TALK-2 channel gating is or in what situations the lower gate plays a role in overall channel regulation. As we have demonstrated, the regulation of TALK-2 activity through the positive gate coupling requires the simultaneous opening of both gates. Thus, the positive coupling potentially enables the integration of intracellular (e.g., oleoyl-CoA, pH_i) and extracellular signals (e.g., pH_e , membrane voltage) targeting either gate, ultimately leading to channel activation. We now have included a section 'Physiological relevance of positive gate coupling in TALK-2' in the discussion that reads as follows:

The existence of two gates that open and close in a strongly-coupled (i.e., concurrently) fashion seems redundant as closure of one gate is sufficient to prevent permeation. However, in TALK-2 channels the two gates have evolved to respond to different stimuli with the SF gate sensing membrane voltage and extracellular pH whereas the lower gate responds to negatively-charged lipids of the cytoplasmic membrane leaflet. Therefore, in order for the individual gate-specific stimuli to become effective strong

gate coupling is mandatory as opening of one gate would be functionally silent if the other gate stays closed. Interestingly, evolution has solved this problem of effective multi-sensory gating for members of the TREK/TRAAK subgroup in a different way as although the various signals (including temperature, voltage, lipids and pH) are sensed by different regions of the channels they finally converge at a single gate being the SF.

In several experiments, Rb⁺ is used to potentiate TALK2 currents through its supposed action on the SF. However, considering the strong reciprocal coupling between the lower gate and the SF, how can Rb⁺ have a further activation effect on TALK2 in conditions where the lower gate is opened (in presence of 2-APB for example – figure 2)?

We thank the reviewer for this valid comment. Although we characterize the gating as strongly-coupled, it should not be considered as 100 %. None of the stimuli described in the manuscript (i.e., lower gate destabilizing mutations or ligands like 2-APB and oleoyl-CoA), which preferentially open the lower gate, can fully activate the selectivity filter. Even the TALK-2 L264A mutation, which destabilizes the lower gate, shows some degree of Rb⁺ activation (**Supplementary Fig. 6b**). Interestingly, in the double mutant consisting of the two g-o-f point mutations L264A and V146A in the lower gate, almost no Rb⁺ activation is left. This observation is consistent with the double mutation exhibiting higher basal activity at rest and a stronger G-V shift compared to the single mutants. Nevertheless, for clarity, we intentionally refrained from including the representation of the double mutation in the manuscript.

In the method section

-p.23, l.2: “Statistical significance between two groups (respective datasets) was validated using an unpaired Student’s t-test or Wilcoxon rank test after f-test application.” The Wilcoxon rank test should be used if data are not normally distributed and do not depend on f-test results. The latter enables to determine if the variance between the 2 groups is equal. If this is the case, a Student’s t-test should be carried out, if not, a Welch’s t-test needs to be used.

We thank the reviewer for spotting that the statistical tests and sequence of testing was formulated in an unfortunate manner in the methods section. We now included a section that reads as follows:

A Kolmogorow-Smirnow test was used to determine whether measurements were normally distributed. Statistical significance between two groups (respective datasets) was validated using an unpaired two-tailed Student’s t-test.

- p.4, l.15: “Furthermore, thus TALK-2 channels are highly and specifically expressed in the human pancreas and are considered as a risk factor for the pathogenesis of type 2 diabetes.” Considering the meaning of the sentence, “thus” should be removed.

- p.5, l.10: “blocker” → “blockers”

- p.5, l.16: “Further, we demonstrate that the lower gate produced a state-dependent blocker pharmacology that is unique in K2P channels.”

Please rephrase the sentence. Maybe “blocker” should simply be removed?

- p.6, l.8: “the mechanisms how they open the ion permeation pathway are currently unknown.”

Please rephrase the sentence. (maybe replace “how” by “through which”?)

- p.7, l.6: “result” → “results”

- p.8, l.29: “spend” → “spent”

- p.8, l.31: “are” → “is”

- p.13, l.9: “Employing a TALK-2 homology models based on the TASK-1 and TASK-2 structures.” “a” should be removed.

- p.15, l.2: “suggest” → “suggests”

- p.15, l.3: “suggest” → “suggests”

- p.15, l.12: “possible” → “possibly”

- p.15, l.25: “blocker” → “blockers”

- p.28, l.21: “similar” → “similarly”

- p.21, l.22: “with” → “where”

- p.22, l.6, l.22: “with” → “where”

We apologize for the mistakes in writing and thank the reviewer for mentioning these points. We corrected all of the suggested edits.

Reviewer #3 (Remarks to the Author):

In this manuscript, Neelsen and colleagues show that the K2P potassium channel TALK-2 contains a cytoplasmic gate, and that this gate and the gate formed by the selectivity filter are positively coupled. The study is extremely well conducted, and the data fully support the conclusion. This is clearly original work but one wonders if the originality is sufficient for it to be published in Nature Communications. The presence of two gates is classic in channels of the different potassium channels subfamilies, with an upper gate formed by the selectivity filter and a lower gate coupled to the cytoplasmic tail in K2P channels or in the bacterial KcsA channel, or to a voltage-sensing membrane domain for voltage-gated potassium channels. Furthermore, these two gates have been shown to be positively coupled in the K2P channel KCNK0 (reference missing in the manuscript: Ben-Abu et al, Nature Structural & Molecular Biology 16, 71–79, 2009) and in KcsA (Heer et al, eLife 6, 2017), and negatively coupled in voltage-gated potassium channels (reference missing in the manuscript: Panyi & Deutsch, J Gen Physiol 129,403-418, 2007). It has been proposed by Ben-Abu and colleagues that this inverse coupling in leak K2P channels and voltage-gated potassium serve electrical signaling, as leak channels have evolved to be predominantly open, whereas voltage-gated channels must inactivate once activated by a depolarization during an action potential.

We agree with the reviewer, the last sentence of the abstract is indeed somewhat misleading as it is well established that there is a sequential negative coupling of the activation gate and the C-type selectivity filter gate (i.e., opening of activation gate causes inactivation of the SF gate - we also included the reference by Panyi & Deutsch, 2007). However, this type of coupling was not addressed in our manuscript but the question whether the SF gate is open or closed (i.e., conductive or non-conductive) when the activation gate is closed. This question has to our knowledge so far only been addressed by MD simulation performed on SF of KcsA in the closed state (Heer et al., 2017). We have changed the sentence in the abstract and point 4 of the synopsis. It reads now:

This concept might extent to other K⁺ channels that contain two gates (e.g., voltage-gated K⁺ channels) for which a coupled and concurrent opening of the two gates has been suggested, but so far not directly demonstrated. (abstract)

AND

The concept of gate coupling in TALK-2 might extent to other K⁺ channels that contain two gates (e.g., voltage-gated K⁺ channels) for which a coupled and concurrent opening of the two gates has been suggested, but so far not directly demonstrated. (synopsis)

Specific comments: Ben-Abu's study should be cited and discussed in the manuscript. Point 4 of the synopsis and the abstract should be corrected because studies on voltage-gated potassium channels show a negative coupling between the two gates. The manuscript would benefit from a discussion of the physiological significance of the TALK-2 gating (which signals are integrated and what is the impact of positive-coupling between the two gates).

We thank the reviewer for the valuable suggestions. We were aware of the Ben-Abu et al. study but decided not to cite the paper for several reasons. The paper reports on an artificial chimera composed

of the voltage sensor domain (VSD) of the *Shaker* Kv channel and the pore domain of the K_{2P} channel *KCNK0*. *KCNK0* is the *Drosophila* homolog to TREK K_{2P} channels. For TREK/TRAAK subgroup members it is now by structural and functional studies well established that they lack a lower gate but instead are exclusively gated at the SF. The Ben-Abu study was published several years before this insight, and therefore the voltage-dependent gating behavior introduced by the VSD probably does not actually reflect the gating of a lower gate but more likely the gating of the SF gate. Therefore, the reported positive coupling of two gates is likely misleading as the chimera likely possess only one functional gate (i.e., the SF). The study also lacks any direct evidence for a functional lower gate such as cysteine modification experiments. For these reasons we actually prefer to not cite the study. Of note, the paper claims that K_{2P} channels can be considered as always open leak channels, and this view is far from being correct. The P_o of TALK-2 channels reported here for instance was measured around 4 %.

With regard to the second point, we have now added a new section to the discussion, as suggested by the reviewer, which reads as follows:

Physiological relevance of positive gate coupling in TALK-2

The existence of two gates that open and close in a strongly-coupled (i.e., concurrently) fashion seems redundant as closure of one gate is sufficient to prevent permeation. However, in TALK-2 channels the two gates have evolved to respond to different stimuli with the SF gate sensing membrane voltage and extracellular pH whereas the lower gate responds to negatively-charged lipids of the cytoplasmic membrane leaflet. Therefore, in order for the individual gate-specific stimuli to become effective strong gate coupling is mandatory as opening of one gate would be functionally silent if the other gate stays closed. Interestingly, evolution has solved this problem of effective multi-sensory gating for members of the TREK/TRAAK subgroup in a different way as although the various signals (including temperature, voltage, lipids and pH) are sensed by different regions of the channels they finally converge at a single gate being the SF.

REVIEWERS' COMMENTS

Reviewer #1 (Remarks to the Author):

The authors have thoroughly addressed all our comments and have made the necessary modifications to the text accordingly. These clarifications have significantly enhanced and solidified the quality of this excellent study. The paper is now clear, comprehensive, and accessible to the broad readership of Nature Communications.